# Strongly Convex Sets in Riemannian Manifolds

**Damien Scieur**[*]
Samsung AI Lab,
Montreal, Canada.

**David Martínez-Rubio**[*†]
IMDEA Software Institute,
Madrid, Spain.

**Thomas Kerdreux**[*†]
Galeio,
Paris, France.

**Alexandre d'Aspremont**
CNRS & D.I. École Normale Supérieure,
Paris, France.

**Sebastian Pokutta**
Zuse Institute Berlin & Technische Universität,
Berlin, Germany.

## Abstract

Strong convexity plays a key role in designing and analyzing convex optimization algorithms and is well-understood in Hilbert spaces. However, the notion of strongly convex sets beyond Hilbert spaces remains unclear. In this paper, we propose various definitions of strong convexity for uniquely geodesic sets in a Riemannian manifold, examine their relationships, introduce tools to identify geodesically strongly convex sets, and analyze the convergence of optimization algorithms over these sets. In particular, we show that the Riemannian Frank-Wolfe algorithm converges linearly when the Riemannian scaling inequalities hold.

## 1 Introduction

In Euclidean space, *strong convexity* is a key property of convex objects (e.g., normed spaces, functions, or sets). Algorithms exploiting strong convexity enjoy faster convergence, tighter generalization bounds, and stronger concentration properties. For example, strong convexity sharpens concentration inequalities for martingales (Pisier, 1975; 2011), enables faster projection-free optimization methods such as Frank-Wolfe (Demyanov and Rubinov, 1970; Garber and Hazan, 2015; Molinaro, 2020; Kerdreux et al., 2021b;c), and plays a key role in learning theory (Huang et al., 2017; Molinaro, 2020; Kerdreux et al., 2021a; Dekel et al., 2017; Bhaskara et al., 2020a;b) and bandit problems (Bubeck et al., 2018; Kerdreux et al., 2021d).

However, many modern machine-learning problems are posed in *non-Euclidean* settings such as metric spaces or Riemannian manifolds. Examples include optimal transport, where entropy-regularized formulations like Sinkhorn divergences (Cuturi, 2013; Feydy et al., 2019) exploit Wasserstein geometry (Villani, 2009), as well as Wasserstein barycenters, distributions on curved spaces, or manifold-valued data. Such applications have spurred the development of Riemannian optimization methods (Sun et al., 2017; Allen-Zhu et al., 2018; Hosseini and Sra, 2017; 2020).

Recent years have seen a growing interest in Riemannian optimization for deep learning. For instance, specialized architectures such as DreamNet leverage Riemannian geometry to learn from symmetric positive definite (SPD) matrices in vision tasks (Wang et al., 2022a), while robotics applications exploit manifold-aware optimization to acquire complex motor skills (Wang et al., 2022b). Beyond applications, there is also progress in developing learning-to-optimize frameworks on manifolds (Gao et al., 2022) and in designing Riemannian counterparts of neural network architectures, such as residual networks (Katsman et al., 2023). This trend has motivated comprehensive surveys (Fei et al., 2025) highlighting the relevance of geometric optimization in modern machine learning.

Yet, the role of strong convexity in non-Euclidean settings remains poorly understood:

> *What does it mean for a set to be strongly convex in curved spaces,*
> *and how can this structure be exploited algorithmically?*

---

[*]Equal contribution (randomized order)

[†]Work conducted while the author was affiliated with the Zuse Institute Berlin (ZIB).

**Related Work.** While strong convexity of functions extends relatively smoothly to metric and geodesic settings, **the situation for sets is far less clear**. Beyond Euclidean spaces, no single accepted definition exists. Several alternatives have been proposed in metric spaces, particularly in non-positively curved (CAT(0)) spaces (Aleksandrov, 1957), and related ideas have appeared in concentration inequalities (Gouic et al., 2019; Ahidar-Coutrix et al., 2020; Mérigot et al., 2020) and in online learning (Paris, 2020; 2021). However, these notions are not equivalent and often diverge in their implications, falling short of providing the structural properties required by modern optimization algorithms.

While (Aleksandrov, 1957) and subsequent CAT(0) literature characterize curvature-based convexity of the ambient space, our work can be viewed as a set-level refinement of that idea. In CAT(0) spaces, convexity follows globally from nonpositive curvature; in our framework, we analyze how local curvature and the exponential map jointly induce strong convexity of specific subsets. Likewise, (Paris, 2020; 2021) leverages geodesic convexity in online learning but does not quantify set curvature or provide conditions for algorithmic strong convexity. Our definitions, therefore, bridge these directions: they recover metric convexity when curvature is nonpositive, while additionally providing geometric and algorithmic tools that apply beyond CAT(0) settings.

Finally, the Riemannian scaling inequality in Eq. (RSI) is conceptually related to proximal smoothness (Davis et al., 2025), yet the two notions are analytically opposite. Proximal smoothness (and its uniform normal inequality) requires an upper bound that holds uniformly over all outward tangent directions. In contrast, RSI provides a one-sided lower bound evaluated only at the LMO maximizer within a restricted tangent subset. The two conditions are thus dual in structure and tailored to distinct algorithmic regimes.

**Contributions.** This paper provides the first systematic study of strong convexity of sets in Riemannian manifolds, bridging geometric structure and algorithmic benefits. We introduce novel properties of sets defined in Riemannian manifolds, provide examples, and demonstrate improved convergence rates when minimizing functions over such sets. For conciseness, proofs are deferred to Section A and technical results to Section B.

1. Section 2 introduces various definitions of strongly convex sets in Riemannian manifolds and establishes relationships between them.
2. Section 3 introduces the notion of approximate scaling inequality and its link with what we call *double geodesic strong convexity*.
3. In Section 4, we prove that sublevel sets of geodesically smooth, strongly convex functions are strongly convex in Riemannian manifolds under mild conditions, providing a constructive perspective for identifying such sets.
4. In Section 5, we derive a linear convergence bound for the Riemannian Frank–Wolfe algorithm when the constraint set satisfies the (Approximate) Riemannian Scaling Inequality.
5. Sections D and E provide examples of strong convex sets in Riemannian manifolds. More precisely, Section D shows that Riemannian balls (with restricted radius) satisfy the strongest of our notions of Riemannian strong geodesic convexity. Section E details a simple linear minimization oracle for the sphere manifold.

Although strong convexity can be verified directly using their definitions, our results provide more practical, Euclidean-style certification tools. In particular, (i) Geodesic balls with bounded radius are Riemannian strongly convex (Theorem D.4) and (ii) Sublevel sets of smooth, strongly convex functions are both geodesically and double-geodesically strongly convex (Theorem 4.2).

**Impact.** Defining and identifying strongly convex sets on manifolds provides the necessary foundation to design algorithms that can exploit this structure. For example, once such sets are characterized, the Riemannian Frank–Wolfe (RFW) achieves faster convergence, analogous to the Euclidean setting (Garber and Hazan, 2015; Braun et al., 2022). Extending these ideas to curved spaces thus offers both theoretical and practical tools for scalable Riemannian optimization in machine learning.

PRELIMINARIES, NOTATIONS, AND ASSUMPTIONS

**Strong Convexity in Hilbert Spaces** In Hilbert spaces, there exist many equivalent definitions of strongly convex sets (Goncharov and Ivanov, 2017). We recall here two classical definitions.

**Proposition 1.1** (Strong Convexity of Sets in Hilbert Spaces (Goncharov and Ivanov, 2017))**.** *Consider a compact convex set $\mathcal{C} \subset \mathbb{R}^n$, its boundary $\partial\mathcal{C}$, $\alpha > 0$, and a norm $\|x\| = \sqrt{\langle x, x \rangle}$. $\mathcal{C}$ is $\alpha$-strongly convex w.r.t. $\|\cdot\|$ if and only if it satisfies the following equivalent assertions.*
*(a) For all $(x, y) \in \mathcal{C} \times \mathcal{C}$, $t \in [0, 1]$, and $z \in \mathbb{R}^n$ s.t. $\|z\| = 1$, we have*

$$(1 - t)x + ty + \alpha(1 - t)t\|x - y\|^2 z \in \mathcal{C}. \tag{1}$$

*Intuitively, a set is called strongly convex if, for all straight lines in the set and all points $p$ in the line, there exists a ball around $p$ of a certain radius that is contained in the set.*
*(b) For all $(x, v) \in \mathcal{C} \times \partial\mathcal{C}$ and $w \in N_{\mathcal{C}}(v)$ (the normal cone of $\mathcal{C}$ at $v$), we have*

$$\langle w; v - x \rangle \geq \alpha\|w\|\|v - x\|^2. \tag{scaling ineq.}$$

In the original definitions, $\alpha$ takes the value $1/2R$, where $R > 0$ is the radius of some balls that are used for an alternative equivalent definition of strong convexity of sets (Vial, 1982; 1983; Goncharov and Ivanov, 2017). We use $\alpha$ in this work since it is simpler for our purposes. We also note that (scaling ineq.) is equivalent to the following condition (Kerdreux et al., 2021b, Lemma 2.1):

$$\forall (x, w) \in \mathcal{C} \times \mathbb{R}^n, \quad \langle w; v - x \rangle \geq \alpha\|w\|\|v - x\|^2 \quad \text{where} \quad v \in \underset{z \in \mathcal{C}}{\arg\max}\langle w; z \rangle. \tag{2}$$

Moreover, Journée et al. (2010, Theorem 12) shows that the sublevel sets of $L$-smooth, $\mu$-strongly convex functions (see Definition C.17) are themselves strongly convex. In particular, the following set is $\mu/(2\sqrt{2Ls})$-strongly convex:

$$Q_s := \{x \mid f(x) - f^* \leq s\}, \quad \text{where} \quad f^* = \min_x f(x). \tag{3}$$

**Notations** We now summarize the essential notations. The reader can find in Section C notions about metric spaces (Section C.1), Riemannian manifolds (Section C.2), sets defined in Riemannian manifolds (Section C.3), and functions defined over a manifold (Section C.4).

**Metric and Geodesic Spaces.** A *metric space* is a pair $(\mathcal{M}, d_{\mathcal{M}})$, where $\mathcal{M}$ is a set and $d_{\mathcal{M}}(x, y)$ is a distance function between $x, y \in \mathcal{M}$ that satisfies positivity, symmetry, and the triangle inequality. A *geodesic* $\gamma : [0, 1] \to \mathcal{M}$ between $x$ and $y$ is a curve with no intrinsic acceleration satisfying $\gamma(0) = x$, $\gamma(1) = y$. If any two points in $\mathcal{M}$ can be connected by such a geodesic, it is a *geodesic metric space*; if the geodesic is unique, it is *uniquely geodesic*.

**Riemannian Manifolds.** $\mathcal{M}$ is a Riemannian manifold if it is equipped with a *Riemannian metric* $g$, which defines for all $x \in \mathcal{M}$ an inner product $\langle \cdot, \cdot \rangle_x$ on the tangent space $T_x\mathcal{M}$. The distance $d_{\mathcal{M}}(x, y)$ is given by the length of the shortest geodesic connecting $x$ and $y$. For all $x \in \mathcal{M}$ and $v \in T_x\mathcal{M}$, let $\gamma : [0, 1] \to \mathcal{M}$ be the geodesic satisfying $\gamma(0) = x$ with $\gamma'(0) = v$. The *exponential map* $\text{Exp}_x : T_x\mathcal{M} \to \mathcal{M}$, is defined as $\text{Exp}_x(v) := \gamma(1)$. *Cartan-Hadamard* spaces (*a.k.a.* complete CAT(0) spaces) are complete, simply connected Riemannian manifolds with non-positive curvature (Definition C.4).

**Main assumptions.** In this paper, we will assume the following statements.

**Assumption 1.2.** *The set $\mathcal{C} \subset \mathcal{M}$ is compact and **uniquely geodesically convex**; i.e., for any $x, y \in \mathcal{C}$, there exists a unique minimizing geodesic contained entirely in $\mathcal{C}$.*

**Assumption 1.3** (Distances Equivalence)**.** *Let the set $\mathcal{C} \subset \mathcal{M}$ satisfies Assumption 1.2 and let $d : \mathcal{C} \times \mathcal{C} \to \mathbb{R}^+$ be a distance function, possibly different from the Riemannian distance. There exists $0 < l_{\mathcal{M}} \leq L_{\mathcal{M}}$ such that for all pairs $(x, y) \in \mathcal{C}^2$,*

$$\ell_{\mathcal{M}}\|\text{Exp}_x^{-1}(y)\|_x \leq d(x, y) \leq L_{\mathcal{M}}\|\text{Exp}_x^{-1}(y)\|_x. \tag{4}$$

Since we assume $\mathcal{C}$ to be uniquely geodesic, the inverse exponential map $\text{Exp}_x^{-1}(y)$, also called logarithmic map, is well defined (see Remark 1 in Section C.3 for more details).

**Discussion** Assumption 1.2 is standard and essentially requires that the exponential map is bijective. This condition is needed in many optimization settings; for example, the RFW algorithm relies on the inverse exponential map (Weber and Sra, 2023). Moreover, Assumption 1.3 becomes non-restrictive under Assumption 1.2: if the exponential map is bijective, the metric distance can be taken to coincide with the norm of the logarithmic map, yielding $\ell_\mathcal{M} L_\mathcal{M} = 1$ (see Remark 1 in Section C.3). We retain this assumption only to cover cases where an alternative distance $d$ is used to define strong convexity or smoothness of a function $f$ defined on $\mathcal{M}$.

## 2 STRONGLY CONVEX SETS IN RIEMANNIAN MANIFOLDS

|  | Geodesic str. cvx | Double geodesic str. cvx | Riemannian str. cvx |
|---|---|---|---|
| **Tangent** | No | Yes | Yes |
| **Exp/Log** | No | Exp + Parallel Transport | Log map |
| **Meaning** | Exist a geodesic ball around geodesic points | Exist a geodesic ball in the tangent space | Log-image of the set is Euclidean str. cvx. |
| **Example** | Same as double geodesic str cvx | Sub-level sets of smooth, strongly convex functions | Small balls in manifolds with bounded-curvature |

Table 1: Summary of the three definitions of strong convexity introduced in this paper

We propose several definitions of strongly convex sets for Riemannian manifolds, which all collapse to Eq. (1) when $\mathcal{M}$ is the Euclidean space (see Table 1 for a summary).

**Geodesic Strong Convexity** The *geodesic strong convexity* adapts (1) using geodesics in $\mathcal{M}$, without considering its Riemannian structure (i.e., the definition does not use the tangent space).

**Definition 2.1** (Geodesic Strong Convexity). *Let $\mathcal{M}$ be a Riemannian manifold, and let $\mathcal{C} \subseteq \mathcal{M}$ satisfies Assumption 1.2. The set $\mathcal{C}$ is geodesically $\alpha$-strongly convex w.r.t. the distance function $d : \mathcal{M} \times \mathcal{M} \to \mathbb{R}^+$ if, for all geodesics $\gamma$ joining $x, y \in \mathcal{C}$ and every $t \in [0, 1]$, we have that the following ball is in $\mathcal{C}$:*

$$\{z \in \mathcal{M} \mid d(\gamma(t), z) \leq \alpha t(1-t)d^2(x, y)\} \subseteq \mathcal{C}. \tag{5}$$

**Riemannian Strong Convexity** The following definition leverages the Riemannian structure of $\mathcal{M}$ via an assumption on the exponential map. The definition states that the inverse image of the set $\mathcal{C} \subset \mathcal{M}$ by the inverse exponential map at each $x \in \mathcal{M}$ must be strongly convex in $T_x\mathcal{M}$ for all $x$ in the Euclidean sense. For a uniquely geodesic set $\mathcal{C}$, the inverse exponential map is always well defined for any two points in $\mathcal{C}$, cf. Remark 1.

**Definition 2.2** (Riemannian Strong Convexity). *Let $\mathcal{M}$ be a Riemannian manifold, and $\mathcal{C} \subseteq \mathcal{M}$ satisfies Assumption 1.2. The set $\mathcal{C}$ is a Riemannian $\alpha$-strongly convex set if, for all $x \in \mathcal{C}$,*

$$\mathrm{Exp}_x^{-1}(\mathcal{C}) := \{y \in T_x\mathcal{M} : y = \mathrm{Exp}_x^{-1}(z), \ z \in \mathcal{C}\}$$

*is $\alpha$-strongly convex w.r.t. $\|\cdot\|_x$ in the Euclidean sense (1).*

**Double Geodesic Strong Convexity** In Definition 2.3, we now leverage the Riemannian structure through the existence of the tangent space and the exponential map to provide another notion of strong convexity of a set, analogous to the Euclidean formulation in (1). For $t \in [0, 1]$, we make the parallel between the term $tx + (1-t)y$ in (1) and $\gamma(t)$, the geodesic $\gamma$ joining $y$, and $x$ in $\mathcal{M}$. Then, (1) in $\mathcal{M}$ ensures that we have $\mathrm{Exp}_{\gamma(t)}\left(\alpha t(1-t)d^2(x, y)z\right) \in \mathcal{C}$ for all $z \in T_{\gamma(t)}\mathcal{M}$ with $\|z\| = 1$.

**Definition 2.3** (Double Geodesic Strong Convexity). *Let $\mathcal{M}$ be a Riemannian manifold, and $\mathcal{C} \subseteq \mathcal{M}$ satisfies Assumption 1.2. The set $\mathcal{C}$ is a double geodesically $\alpha$-strongly convex set w.r.t. $d(\cdot, \cdot)$ if, for all geodesics $\gamma$ joining $x, y \in \mathcal{C}$ and for all $t \in [0, 1]$,*

$$\forall z \in T_{\gamma(t)}\mathcal{M} : \|z\|_{\gamma(t)} \leq \alpha t(1-t)d^2(x, y) \quad \Rightarrow \quad \mathrm{Exp}_{\gamma(t)}(z) \text{ exists and is in } \mathcal{C}. \tag{6}$$

The double geodesic strong convexity can also be rewritten in terms of the exponential map if $\mathrm{Exp}_{\gamma(t)}(\alpha t(1-t)d^2(x, y)z) \in \mathcal{C}$ for all $z$ of unit norm in $T_{\gamma(t)}\mathcal{M}$. In this manner, it mirrors the algebraic expression of the Euclidean definition we provided in (1) but within the Riemannian setup. We

Figure 1: Illustration of the different definitions of strong convexity.
**(Left)** Geodesic strong convexity (Definition 2.1): For each pair of points $x, y$ in the set $\mathcal{C}$, it is possible to draw a "geodesic ball" centered at a point on the geodesic $\gamma(t)$ connecting $x$ and $y$, such that the ball remains entirely within $\mathcal{C}$. **(Center)** Riemannian strong convexity (Definition 2.2): For all $x \in \mathcal{C}$, the image of the set under the inverse exponential map, $\mathrm{Exp}_x^{-1}(\mathcal{C})$, is a strongly convex set. **(Right)** Double geodesic strong convexity (Definition 2.3): The image of a ball in $T_{\gamma(t)}\mathcal{M}$ of radius $\alpha t(1-t)d^2(x, y)$ under the exponential map $\mathrm{Exp}_{\gamma(t)}(\cdot)$ is contained within $\mathcal{C}$.

denote it as the *double geodesic strong convexity* also because the point $\mathrm{Exp}_{\gamma(t)}(\alpha t(1-t)d^2(x, y)z)$ is built via two geodesics, one between $x$ and $y$, and another starting at $\gamma(t)$.

**Riemannian Scaling Inequality (RSI)**    In Euclidean space, (scaling ineq.) is an equivalent definition of a strongly convex set (Prop. 1.1), which helps in establishing convergence proofs of various algorithms. Here, we propose the notion of the RSI, which is the the Riemannian counterpart of (2).

**Definition 2.4** (Riemannian Scaling Inequality (RSI)). *Let $\mathcal{M}$ be a Riemannian manifold, and let $\mathcal{C} \subset \mathcal{M}$ satisfies Assumption 1.2. The elements in the set $\mathcal{C}$ then satisfy the RSI if, for some $\alpha > 0$, for all $x \in \mathcal{C}$, $w \in T_x\mathcal{M}$,*

$$\langle w; \mathrm{Exp}_x^{-1}(v)\rangle_x \geq \alpha\|w\|_x\|\mathrm{Exp}_x^{-1}(v)\|_x^2, \quad \textit{for all} \ \ v \in \mathrm{argmax}_{z \in \mathcal{C}}\langle w; \mathrm{Exp}_x^{-1}(z)\rangle_x. \quad \text{(RSI)}$$

We now show some implications and equivalences between strong convexity notions and scaling inequality for manifolds. We summarize the links between these notions here.

$$
\begin{array}{l}
\text{Riemannian} \xRightarrow[\text{Hadamard}]{\text{Prop. 2.6,}} \text{Geodesic Strong} \xLeftrightarrow[\text{A. 1.3}]{\text{Prop. 2.7}} \text{Double Geodesic} \xRightarrow{\text{Prop. 3.3}} \text{Approx. RSI} \\
\text{Strong Convexity} \qquad\qquad\quad \text{Convexity} \qquad\qquad\qquad \text{Strong Convexity} \\[2mm]
\text{Riemannian} \xRightarrow{\text{Prop. 2.5}} \text{RSI} \xRightarrow{\phantom{xx}} \text{Approx. RSI} \\
\text{Strong Convexity}
\end{array}
\qquad (7)
$$

In Hilbert spaces, the scaling inequality and global strong convexity of a set are equivalent notions (Proposition 1.1), but not in Riemannian manifolds. Instead, Proposition 2.5 states that the Riemannian strong convexity implies a RSI, the latter being valuable for analyzing algorithms (Section 5).

**Proposition 2.5** (Riemannian Strong Convexity implies RSI). *Let $\mathcal{M}$ be a Riemannian manifold, and $\mathcal{C} \subset \mathcal{M}$ satisfies Assumption 1.2. Let $\mathcal{C}$ be a Riemannian $\alpha$-strongly convex set (Definition 2.2). Then, the elements in the set $\mathcal{C}$ also satisfy the RSI (Definition 2.4).*

**Proposition 2.6** (Riemannian Strong Convexity implies Geodesic Strong Convexity). *Let $\mathcal{M}$ be a Riemannian manifold, and $\mathcal{C} \subset \mathcal{M}$ satisfies Assumption 1.2 and $d(\cdot, \cdot)$ satisfies Assumption 1.3. Let $\mathcal{C}$ be a Riemannian $\alpha$-strongly convex set (Definition 2.2). Then, the set $\mathcal{C}$ is a geodesic $(\alpha \ell_{\mathcal{M}}/L_{\mathcal{M}}^2)$-strongly convex set (Definition 2.1).*

By the same arguments presented in Proposition 2.6, the Riemannian strong convexity implies the geodesic strong convexity of sets in Cartan-Hadamard manifolds. This situation bears resemblance to the various notions of geodesically convex sets for Cartan-Hadamard manifolds (Boumal, 2020). It should be noted that geodesic and double geodesic strong convexity (Definitions 2.1 and 2.3) become equivalent under mild assumptions, which is noteworthy as Definition 2.1 relies on the geodesic metric space structure of $\mathcal{M}$, while Definition 2.3 leverages the manifold structure of $\mathcal{M}$.

**Proposition 2.7** (Equivalence between Geodesic and Double Geodesic Strong Convexity). *Let $\mathcal{M}$ be a Riemannian manifold, and $\mathcal{C} \subset \mathcal{M}$ satisfies Assumption 1.2 and $d(\cdot, \cdot)$ satisfies Assumption 1.3. If $\mathcal{C}$ is a geodesic $\alpha$-strongly convex set, then it is a double geodesic $\frac{\alpha}{L_{\mathcal{M}}}$-strongly convex set. If the set $\mathcal{C}$ is a double geodesic $\alpha$-strongly convex set, then it is a geodesic $\ell_{\mathcal{M}}\alpha$-strongly convex set.*

**Examples of Riemannian Strongly Convex Sets**   As shown in Eq. (7), Riemannian strongly convex sets are the most restrictive case. For such sets $\mathcal{C}$, the logarithmic image $\mathrm{Exp}_x^{-1}(\mathcal{C})$ is Euclidean strongly convex. Intuitively, this occurs when the logarithmic map $\mathrm{Exp}_x^{-1}$ only slightly distorts $\mathcal{C}$, which happens if the manifold $\mathcal{M}$ has bounded sectional curvature and $\mathcal{C}$ is not too large. In particular, on manifolds with bounded curvature, Theorem D.4 in Section D shows that balls with radius below a curvature-dependent bound are Riemannian strongly convex. This theorem applies to most practical cases, such as (but not limited to) spheres, hyperbolic spaces, Grassmann and Stiefel manifolds, the special orthogonal group, and the cone of symmetric positive definite matrices.

**Examples of Geodesic Strongly Convex and Double Geodesic Strongly Convex Sets**   As shown in Theorem 4.2 (see Section 4), level sets of smooth, strongly convex functions on Riemannian manifolds are strongly convex. This extends the Euclidean result of Journée et al. (2010, Thm 12).

## 3   Double Geodesic Strong Convexity and Approximate RSI

Since there is no apparent link between (double) geodesic strong convexity (Definition 2.2, 2.3) and the RSI, this section explore the link between those. We introduce the notion of *approximate RSI* (Definition 3.2) and demonstrate that the quality of the approximation depends on the *exponential map operator* (Definition 3.1). Those results will be useful to prove that the Riemannian FW algorithm (Algorithm 1) admits a global linear convergence rate under certain conditions.

### 3.1   Double Geodesic Strong Convexity and Double Exponential Map

We first introduce the *double exponential map* (Gavrilov, 2007; Dzhepko and Nikonorov, 2008; Nikonorov, 2013) and rewrite the double geodesic strong convexity with two geodesic paths.

**Definition 3.1** (Double Exponential Map). *Let $\mathcal{M}$ be a complete, connected Riemannian manifold. Let $T_x\mathcal{M}$ be the tangent space to $\mathcal{M}$ at $x \in \mathcal{M}$, $\mathrm{Exp}_x(\cdot) : T_x\mathcal{M} \to \mathcal{M}$ be the exponential map at $x$, and $\Gamma_x^y : T_x\mathcal{M} \to T_y\mathcal{M}$ be the transportation map between $T_x\mathcal{M}$ and $T_y\mathcal{M}$. We define the double exponential map at $x \in \mathcal{M}$ as the function $\mathrm{Exp}_x(u,v) : T_x\mathcal{M} \times T_x\mathcal{M} \to \mathcal{M}$, such that*

$$\mathrm{Exp}_x(u,v) := \mathrm{Exp}_{\mathrm{Exp}_x(u)}\left(\Gamma_x^{\mathrm{Exp}_x(u)} v\right). \qquad \text{(double exponential map)}$$

*We also define $h_x(\cdot, \cdot) : T_x\mathcal{M} \times T_x\mathcal{M} \to T_x\mathcal{M}$ as the (unique) exponential map operator, such that*

$$\mathrm{Exp}_x(h_x(u,v)) := \mathrm{Exp}_x(u,v), \;\; \forall u,v \in T_x\mathcal{M}. \qquad \text{(Exponential Map Operator)}$$

In particular, we can rewrite the double geodesic strong convexity of a set with the double exponential map. Informally, Definition 2.3 takes into consideration the geodesic $\gamma$ between $x$ and $y$ and other geodesics departing from a $\gamma(t)$ that moves in every $z$ direction, thus describing a closed ball in the tangent space $T_{\gamma(t)}\mathcal{M}$. Therefore, (6) in Definition 2.3 becomes

$$\mathrm{Exp}_x\left(\mathrm{Exp}_x^{-1}(\gamma(t)), \alpha t(1-t)d^2(x,y)z\right) \in \mathcal{C}, \text{ for all } z \in \bar{B}(0,1) \subset T_x\mathcal{M}. \qquad (8)$$

This expression motivates the use of the term *double* to describe the notion of strong convexity.

### 3.2   Approximate RSI and its Link with Double Geodesic Strong Convexity

We now link the double geodesic strong convexity (Definition 2.3) with an approximate version of the scaling inequality (see Definition 3.2). When the exponential map operator satisfies $h(u,v) = u + v$ (basically, when the set is Euclidean), the double geodesic set strong convexity implies the RSI (Proposition B.1). Instead, in (Dzhepko and Nikonorov, 2008; Nikonorov, 2013), explicit approximations of the *exponential map operator* were proposed when the Riemannian manifold is symmetric or has a constant curvature, e.g., the Euclidean sphere or Lobachevsky spaces (Dzhepko and Nikonorov, 2008). These approximations provide expansions of $h_x(u,v)$ of the form

$$h_x(u,v) = u + v + R_x(u,v), \qquad (9)$$

where the term $R_x(u,v)$ gives the quality of the approximation of $h_x(u,v)$ by $u + v$. The general expression of $R_x(u,v)$ as a series is given in (Gavrilov, 2007). Dzhepko and Nikonorov (2008) provide the explicit Taylor series for the Euclidean sphere (Dzhepko and Nikonorov, 2008, (6)) and Lobachevskii spaces (Dzhepko and Nikonorov, 2008, Section 3).

**Definition 3.2** (Approximate RSI). *Let $\mathcal{M}$ be a Riemannian manifold, and $\mathcal{C} \subset \mathcal{M}$ be a geodesically $\alpha$-strongly convex set (Definition 2.1). We then say that $\mathcal{C}$ satisfies the approximate RSI w.r.t. the distance $d(\cdot, \cdot)$ and the residual $r(\cdot) : \mathcal{C} \to T\mathcal{M}$ if, for all $x \in \mathcal{C}$, $w \in T_x\mathcal{M}$, and $v \in \mathrm{argmax}_{z \in \mathcal{C}} \langle w; \mathrm{Exp}_x^{-1}(z) \rangle_x$,*

$$\langle w; \mathrm{Exp}_x^{-1}(v) \rangle_x \geq \alpha \|w\|_x d(v, x)^2 + \langle w; r(x) \rangle_x, \qquad \text{(Approx. RSI)}$$

Proposition 3.3 demonstrates that the double geodesic strong convexity implies an approximate RSI.

**Proposition 3.3** (Double Geodesic Str. Cvx. implies Approximate RSI). *Let $\mathcal{M}$ be a Riemannian manifold, and $\mathcal{C} \subset \mathcal{M}$ be a double geodesically $\alpha$-strongly convex (Definition 2.3). We define $R_x : T_x\mathcal{M} \times T_x\mathcal{M} \to T_x\mathcal{M}$, such that, for all $x \in \mathcal{C}$, the exponential map operator (Definition 3.1) is decomposed as*

$$h_x(u, v) = u + v + R_x(u, v), \quad \forall (u, v) \in T_x\mathcal{M}, \ x \in \mathcal{C}. \qquad (10)$$

*Then, the approximate RSI (Definition 3.2) is satisfied w.r.t. $d(\cdot, \cdot)$ with the residual $r(\cdot)$ s.t. for all $x \in \mathcal{C}$ and $w \in T_x\mathcal{M}$,*

$$r(x) = R_x \left[ \frac{1}{2} \gamma'_{x,v}(0), \Gamma^x_{\gamma_{x,v}(1/2)} \left( \frac{\alpha}{4} d^2(x, v) z^* \right) \right], \qquad \text{(Residual)}$$

*where $v \in \mathrm{argmax}_{z \in \mathcal{C}} \langle w; \mathrm{Exp}_x^{-1}(z) \rangle_x$ and $z^* \in \mathrm{argmax}_{\|z\|_{\gamma_{x,v}(1/2)} = 1} \left\langle \Gamma^{\gamma_{x,v}(1/2)}_x w; z \right\rangle_{\gamma_{x,v}(1/2)}$.*

The approximate scaling inequality is meaningful in the regime where $x$ and $v$ are close. In this situation, the scale of the residual $r(\cdot)$ is determined by $\gamma'_{x,v}(0)$ and $d^2(x, v)$. In some setting, the residual is such that the approximate term $\langle w; r(x) \rangle_x$ in Definition 3.2 becomes negligible w.r.t. the original term $\alpha \|w\|_x d(x, v)^2$. It should be noted that, in the analysis of the FW algorithm in Section 5, we select $w := -\nabla f(x_t)$, where $x_t$ are the FW iterates.

## 4 SUBLEVEL SETS OF GEODESICALLY STRONGLY CONVEX FUNCTIONS

Proving that a set is strongly convex might be difficult in practice. Hence, we present the Riemannian counterpart of Eq. (3) for sublevel sets of geodesically smooth and strongly convex functions (Definition C.18). These findings extend to the notion of geodesically convex functions (Rapcsák, 2013) and (Boumal, 2020, Proposition 11.8). This result relies heavily on the following lemma.

**Lemma 4.1** (Smoothness property). *Let $f$ be a geodesically $L$-smooth function (Definitions C.18 and C.19) defined on the geodesically closed subset $\mathcal{C} \subset \mathcal{M}$, where $\mathcal{M}$ is an Cartan-Hadamard manifold. We denote $x^* \in \mathrm{argmin}_{x \in \mathcal{C}} f(x)$. Then,*

$$\|\nabla f(x)\|_x \leq \sqrt{2L(f(x) - f(x^*))}. \qquad (11)$$

This result is based on the concept of functional duality in a Riemannian manifold, which has been comprehensively studied in (Bergmann et al., 2021). The proof is deferred in Section B.2.

**Theorem 4.2** (Geodesic Strong Convexity of Sublevel Sets). *Let $\mathcal{M}$ be a Riemannian manifold, and $\mathcal{C} \subset \mathcal{M}$ satisfies Assumption 1.2. Let $f : \mathcal{C} \subseteq \mathcal{M} \to \mathbb{R}$ is a proper, geodesically $L$-smooth, and $\mu$-strongly convex function on $\mathcal{C}$ w.r.t. the distance function $d$ satisfying Assumption 1.3. Let $x^* \in \mathcal{C}$ satisfying $\nabla f(x^*) = 0$. Let $Q_s := \{ x \mid f(x) - f^* \leq s \} \subseteq \mathcal{C}$ be geodesically strictly convex for some $s > 0$, that is, every geodesic segment in $Q_s$ is in the interior of $Q_s$ except possibly for its endpoints. Then, $Q_s$ is a geodesically strongly convex set with $\alpha = \mu / 2(2sL \max\{\ell_{\mathcal{M}}^{-2}; 1\})^{1/2}$.*

**Example: Unit sphere.** Let us consider $\mathbb{S}^{n-1}$, the unit sphere manifold embedded in $\mathbb{R}^n$, with the distance function $d(x, y) = \arccos(\langle x; y \rangle)$. Let us fix $x_0 \in \mathcal{M}$, and let $f(x) = d^2(x_0, x)$. Let the set $Q_s := \{ x : f(x) \leq s \}$. When $s < \left( \frac{\pi}{2} \right)^2$, the squared distance function is a geodesically smooth and strongly convex function (the constants of which depend on $s$) (Lee, 2018, Lem. 12.15), (Sakai, 1996, pp153–154). As $Q_s$ is also a strictly convex set for $s < (\pi/2)^2$, the set $Q_s$ is a geodesically strongly convex set, as shown in Theorem 4.2.

**Example: Symmetric Positive Definite Matrices.** The manifold $\mathcal{M}$ of positive definite matrices with the distance function $d_{\mathcal{M}}(X, Y) = (\sum_i \log^2 \lambda_i(X^{-1}Y))^{1/2}$ is a Cartan-Hadamard manifold (Hosseini and Sra, 2015). Let us fix $X_0 \in \mathcal{M}$, and let $f(X) = d^2_{\mathcal{M}}(X_0, X)$. As $\mathcal{M}$ is a Cartan-Hadamard manifold, its distance function is strongly convex, and $d$ is also smooth in bounded sets. Therefore, the sets $Q_s$ (for $s < f(0)$) are geodesically strongly convex, as shown in Theorem 4.2.

## 5 FRANK-WOLFE ON GEODESICALLY STRONGLY CONVEX SETS

The FW algorithm is a first-order method that solves constrained optimization problems in Banach spaces. Each iteration relies on a linear minimization step over the constraint region. It has recently been extended for constrained optimization over Riemannian manifolds. The Riemannian Frank-Wolfe (RFW) algorithm (Weber and Sra, 2023; 2019) minimizes smooth convex functions $f$ over compact and geodesically convex constrained sets $\mathcal{C} \subseteq \mathcal{M}$, i.e., it solves

$$\text{minimize } f(x), \quad \text{for } x \in \mathcal{C}. \tag{OPT}$$

---

**Algorithm 1** Riemannian Frank-Wolfe (RFW) algorithm

---

**Require:** $x_0 \in \mathcal{C} \subset \mathcal{M}$; assume access to the geodesic map $\gamma : [0,1] \to \mathcal{M}$
   **for** $t = 0, 1, \cdots$ **do**
       **1.** $v_t \leftarrow \operatorname{argmin}_{v \in \mathcal{C}} \langle \nabla f(x_t); \operatorname{Exp}_{x_t}^{-1}(v) \rangle_{x_t}$
       **2.** $s_t \leftarrow \operatorname{argmin}_{s \in [0,1]} s \langle \nabla f(x_t); \operatorname{Exp}_{x_t}^{-1}(v_t) \rangle_{x_t} + s^2 \frac{L}{2} d^2(x_t, v_t)$.
       **3.** $x_{t+1} \leftarrow \gamma(s_t)$, where $\gamma(0) = x_t$ and $\gamma(1) = v_t$.
   **end for**

---

Remarkably, (Weber and Sra, 2023; 2019) proves similar convergence rates of RFW as the FW algorithm with comparable assumptions on the optimization problem as in the Hilbertian setting. For instance, when the function $f$ is geodesically convex and smooth and the set is compact convex, the RFW algorithm converges in $\mathcal{O}(1/T)$ (Weber and Sra, 2023, Theorem 3.4.). Similarly, Weber and Sra (2023, Theorem 3.5.) show linear convergence of Algorithm 1 when using short-step sizes, and the objective function is geodesically strongly convex and the optimum in the interior of the set.

In the Hilbertian setting, the FW algorithm admits various accelerated convergence regimes when the set is strongly convex. When the unconstrained optimum of $f$ is outside the constraint set, the FW algorithm converges linearly (Demyanov and Rubinov, 1970). When the function is strongly convex, the convergence rate reads $\mathcal{O}(1/T^2)$ without assumption on the unconstrained optimum location (Garber and Hazan, 2015). The previous sections establish possible notions of strong convexity for sets in Riemannian manifolds. We now demonstrate that analog convergence regimes for the RFW algorithm on geodesically convex sets hold as in the case of the Hilbertian setting. These results complete the work of (Weber and Sra, 2023; 2019).

### 5.1 LINEAR CONVERGENCE OF RFW UNDER RSI

The *scaling inequality* (Proposition 1.1.(b)) is a convenient characterization of the strong convexity of the set (in Hilbertian setting) for establishing convergence rates (Kerdreux et al., 2021b). We follow the same path and establish the convergence rate of the RFW algorithm when the constraint set satisfies (approximate) Riemannian scaling inequalities. In Theorem 5.1, we prove the linear convergence of the RFW algorithm when the set $\mathcal{C}$ satisfies a RSI and the unconstrained optimum of $f$ is outside $\mathcal{C}$. This generalizes (Demyanov and Rubinov, 1970) in a Riemannian setting.

**Theorem 5.1** (Linearly Convergent RFW with RSI). *Consider a geodesically convex set $\mathcal{C}$ and a geodesically convex function $f$ that is $L$-smooth in $\mathcal{C}$. Assume that any unconstrained optimum of $f$ lies outside the constraint set $\mathcal{C}$, and in particular, there exists $c > 0$ s.t. $\operatorname{argmin}_{x \in \mathcal{C}} \|\nabla f(x)\|_x > c$. Assume that, for all $x \in \mathcal{C}$, the (RSI) holds. Then Algorithm 1 converges linearly:*

$$f(x_{t+1}) - f(x^*) \le (f(x_t) - f(x^*)) \max\{1/2, 1 - \alpha c/(2L)\}.$$

### 5.2 LOCAL LINEAR CONVERGENCE OF RFW UNDER THE APPROXIMATE RSI

Since the "exact" RSI could be too restrictive for some applications, this section provides a similar convergence result when the feasible sets satisfy only approximately the RSI (Definition 3.2).

**Theorem 5.2** (Linearly Convergent RFW on Double geodesically strongly convex Sets). *Let $\mathcal{M}$ be a Riemannian manifold, and $\mathcal{C} \subset \mathcal{M}$ be an $\alpha-$double geodesically strong convex (Definition 2.3) set w.r.t. the distance $d(\cdot, \cdot)$ satisfying Assumption 1.3. Assume that the function $f : \mathcal{C} \to \mathbb{R}$ is a geodesically convex $L$-smooth function, and there exists $c > 0$ s.t. $\min_{x \in \mathcal{C}} \|\nabla f(x)\|_x > c$. Assume*

*that there exists some constant $C > 0$ s.t. the residual (10) of $R_x(\cdot, \cdot)$ satisfies*

$$\|R_x(u, w)\|_x \leq C \cdot \max\left\{\|u\|_x^2\|w\|_x; \|w\|_x^2\|u\|_x\right\} \quad \forall x \in \mathcal{C}, \ \forall u, w \in T_x\mathcal{M}. \qquad (12)$$

*Assume that $\exists \delta > 0 : d(x_t, v_t)^2 \leq (\alpha c)/(2\delta L\tilde{C})$, where $\tilde{C} := C \ \max\left\{\frac{\alpha^2}{16\delta\ell_\mathcal{M}}; \frac{\alpha}{4\ell_\mathcal{M}^2}\right\}$. Then,*

$$f(x_{t+1}) - f(x^*) \leq (f(x_t) - f(x^*))\max\left\{\tfrac{1}{2}, 1 - \tfrac{\alpha c}{2L}\right\}.$$

The previous theorem thus states that, provided a burn-in phase (that follows from the general $\mathcal{O}(1/T)$ of the RFW algorithm) to ensure $d(x_t, v_t)^2 \leq (\alpha c)/(2\delta L\tilde{C})$, the iterates of the RFW algorithm converge linearly when the set satisfies the approximate RSI.

**Discussion on the residual bound** When the feasible set is Riemannian strongly convex (see examples in Section D), Eq. (12) is automatically satisfied, so that both the exact and approximate RSI hold. Moreover, the approximate RSI can be satisfied in settings where the exact RSI fails. For instance, Dzhepko and Nikonorov (2008) analyze the double exponential map on spaces of constant curvature, such as spheres and hyperbolic manifolds, and show that the residual admits an expansion of the form of Eq. (12). Importantly, this constant *does not need to be known* in practice, since the RFW algorithm automatically switches to linear convergence once the condition is satisfied.

**Example 1: Riemannian trust-region subproblem.** When minimizing a function $f$ on a Riemannian manifold, a common approach is the trust-region method. After $t$ iterations, it approximates $\tilde{f}_t \approx f$, where $\tilde{f}_t$ is usually a second-order approximation of $f$ around $x_t$. Common trust region approaches minimize $\tilde{f}_t(x_t + \Delta)$, where $\Delta$ belongs to the tangent space of $x_t$ under the constraint that $\|\Delta\| \leq \delta_t$; we then use a retraction on $x_t + \Delta$ to obtain the iterate $x_{t+1}$. Alternatively, it is possible to solve the subproblem directly on the manifold:

$$x_{t+1} = \operatorname{argmin} \tilde{f}_t(x), \qquad \text{subject to } x \in \mathcal{M}, \ d(x_t, x) \leq \delta_t.$$

Using Theorem 4.2, we determine that the set is geodesically strongly convex. Therefore, if the set is sufficiently small, the RFW algorithm converges linearly on this subproblem.

**Example 2: Global Riemannian optimization through local subproblem solving** For manifolds of curvature bounded in $[\kappa_{\min}, \kappa_{\max}]$, and defining $K \overset{\text{def}}{=} \max\{|\kappa_{\min}|, \kappa_{\max}\}$, Martínez-Rubio (2020) presented a reduction from global Riemannian g-convex optimization to optimization in Riemannian balls of radius $O(\frac{1}{\sqrt{K}})$. It is required to solve $O(\zeta_R)$ of such ball optimization problems to an accuracy proportional to the final global accuracy (up to low polynomial factor). Here, $R$ is the initial distance to an optimizer, and $\zeta_R$ is a natural geometric constant. If we solve the subproblems using linear rates, the reduction only adds a $\widetilde{O}(\zeta_R)$ factor to these linear rates.

### 5.3 Numerical Experiment: Minimization Over a Sphere

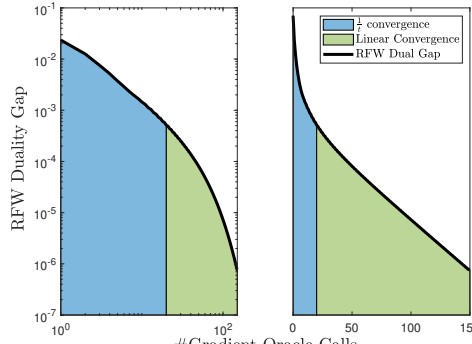

Figure 2: Numerical convergence of the RFW algorithm's iterates (Algo. 1) for minimizing a quadratic function over a double geodesically strongly convex set in a unit sphere. The dimension of the problem is $n = 500$, $x_c$ is a vector of ones, and the function $f$ is a random quadratic function parametrized as $f(x) \triangleq \|A(x - x^\star)\|^2/2$, where $A$ is a random $250 \times 500$ matrix, and $x^\star$ is generated at random such that $\text{dist}(x^\star, x_c) \leq \pi/2$. The parameter $R$ is set such that $R = 0.9\,\text{dist}(x^\star, x_c)$, which ensures that the solution lies on the boundary, and therefore, there exists $c > 0$ s.t. $\operatorname{argmin}_{x \in \mathcal{C}}\|\nabla f(x)\|_x > c$. As predicted by Theorem 5.2, the rate is locally linear.

This section presents a numerical experiment illustrating the rates stated in Theorem 5.2. Let the manifold $\mathcal{M}$ be $\mathbb{S}^{n-1}$, the unit sphere embedded in $\mathbb{R}^n$. Consider the problem

$$\min f(x), \qquad \text{subject to } x \in \mathbb{S}^{n-1}, \ \text{dist}(x, x_c) \leq R < \tfrac{\pi}{2}. \qquad (13)$$

Owing to the symmetries of the sphere, the linear minimization oracle for (13) can be formulated as a simple one-dimensional problem (see Section E and Proposition E.2). The experimental setting and results have been reported in Fig. 2, wherein the two regimes (global rate of $O(t^{-1})$ and locally linearly convergent) are clearly distinguishable. This problem appears, for instance, when training a neural network with spherical constraints over a hierarchical dataset.

**Example 5.3** (Hierarchical NN with sphere constraints). *Scieur and Kim (2021) trained a neural network on a hierarchical dataset: "the classifier of each class belongs to a sphere whose center is the classifier of its super-class". The problem can be formulated as in (13), where $x_c$ is the separating hyperplane of the super-class, and $f$ is the loss over the dataset.*

## 5.4 Example: Finite-temperature free energy minimization

We consider a minimization problem over a geodesic ball in the SPD manifold:

$$\min_{X \in \mathcal{S}_{++}^n} f(X) \qquad \text{s.t.} \qquad d(X, X_0) \leq r.$$

(See Section F.) The following experiment illustrates how the theoretical rates manifest in practice. For a temperature $T > 0$, we seek

$$X^\star \in \arg\min_{0 \preceq X \preceq I} F(X) = E_\mu(X) - T\,S(X), \qquad S(X) = -\text{Tr}[X \log X + (I - X)\log(I - X)],$$

where $E_\mu$ is the electronic energy with chemical potential $\mu$ enforcing $\text{Tr}\,X = N_e$. Here $X$ is the finite-temperature density matrix. In standard electronic-structure computations, this problem is typically approached using the relaxed fixed-point iteration

$$X_{k+1} = (1 - \alpha_k)X_k + \alpha_k\,\Phi(\nabla E(X_k)),$$

where $\alpha_k$ is chosen by a heuristic line search and the map $\Phi$ is computed from the eigen-decomposition $\nabla E(X_k) = U\Lambda U^\top$:

$$\Phi(U\Lambda U^\top) = U \,\text{diag}\big(1 + e^{(\lambda_i - \mu)/T}\big)^{-1} U^\top.$$

Thus $\Phi$ applies the Fermi–Dirac function to the eigenvalues of the Hamiltonian, and the iteration amounts to minimizing a first-order (linearized) model of $E$ at $X_k$. A more controlled alternative is to linearize only the energy around a reference point $X_0$, while keeping the entropy term exact. Let $\nabla E(X_0)$ denote the gradient at $X_0$. We consider the problem

$$\min_{X \in \mathcal{X}} \langle \nabla E(X_0), X - X_0\rangle_X - TS(X) \qquad \text{s.t.} \qquad d(X, X_0) \leq r,$$

with $d(\cdot, \cdot)$ the chosen Riemannian distance and $r > 0$ a trust-region radius. This trust-region subproblem provides a stable local model for updating $X_0$ while explicitly controlling the geometric displacement measured by $d$.

## Conclusion

We presented the first definitions for strong convexity of sets in Riemannian manifolds, studied their relationships, and provided examples of these sets. The global linear convergence of the RFW algorithm serves as a tangible demonstration of the impact of developing a theory around the strong convexity structure of these sets.

We expect strongly convex structure to be helpful when developing Riemannian algorithms in the contexts where the Euclidean algorithm counterpart was leveraging such a structure, e.g., in the generalized power method (Journée et al., 2010), online learning (Huang et al., 2017; Dekel et al., 2017; Bhaskara et al., 2020a;b), and even more broadly, in the case of the use of *strongification* techniques, as in (Molinaro, 2020).

ACKNOWLEDGMENTS

Research reported in this paper was partially supported through the Research Campus Modal funded by the German Federal Ministry of Education and Research (fund numbers 05M14ZAM,05M20ZBM) and the Deutsche Forschungsgemeinschaft (DFG) through the DFG Cluster of Excellence MATH+ (EXC-2046/1 and EXC-2046/2, project id 390685689). David Martínez-Rubio was partially funded by grant PID2024-160448NA-I00 and La Caixa Junior Leader Fellowship 2025. AdA would like to acknowledge support from a Google focused award, as well as funding by the French government under management of Agence Nationale de la Recherche as part of the "Investissements d'avenir" program, reference ANR-19-P3IA-0001 (PRAIRIE 3IA Institute). We would like to thank Editage (www.editage.co.kr) for English language editing.

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

CONTENTS

# A    MISSING PROOFS

**Proof of Proposition 2.5**

*Proof.* As the set is strongly convex in the Euclidean sense, we obtain from (scaling ineq.) that, for all $x \in \mathcal{C}$ and for all $w \in T_x\mathcal{M}$, we have

$$\langle w; u \rangle_x \geq \alpha \|w\|_x \|u - x\|_x^2, \quad u \in \underset{z \in \mathrm{Exp}_x^{-1}(\mathcal{C})}{\mathrm{argmax}} \ \langle w; z \rangle_x.$$

As the exponential map is bijective over the set $\mathcal{C}$,

$$\underset{z \in \mathrm{Exp}_x^{-1}(\mathcal{C})}{\mathrm{argmax}} \ \langle w; z \rangle_x = \mathrm{Exp}_x^{-1}\left( \underset{z \in \mathcal{C}}{\mathrm{argmax}} \langle w; \mathrm{Exp}_x^{-1}(z) \rangle_x \right).$$

Therefore, using $v = \mathrm{Exp}_x(u)$ gives us

$$\langle w; \mathrm{Exp}_x^{-1}(v) \rangle_x \geq \alpha \|w\|_x \|\mathrm{Exp}_x^{-1}(v)\|_x^2, \quad v \in \underset{z \in \mathcal{C}}{\mathrm{argmax}} \langle w; \mathrm{Exp}_x^{-1}(z) \rangle_x.$$

$\square$

**Proof of Proposition 2.6**

*Proof.* First, we introduce the important notion of *geodesic triangle*. For the three points $p, q, r \in \mathcal{M}$, the set $\Delta pqr$ of the three minimizing geodesics joining these three points is a *geodesic triangle*. A *comparison triangle* $\Delta \bar{p}\bar{q}\bar{r}$ is then a triangle with the same side length as $\Delta pqr$ in a metric space with a constant *sectional curvature*. *Comparison theorems* are then used to compare the angle between these triangles according to a lower (Toponogov's theorem) or an upper bound (Rauch's theorem) on the sectional curvature of the geodesic metric space $\mathcal{M}$. We refer to (Burago et al., 1992; Meyer, 1989; Burago et al., 2001) for a detailed treatment of such comparison theorems and to (Zhang and Sra, 2016; 2018) for their use in optimization contexts. It should be noted that comparison theorems do not only compare triangles, but also *hinges*, and result in angle or length comparisons (Meyer, 1989, Theorem 2.2. B).

As the set $\mathcal{C}$ is a Riemannian strongly convex set (Definition 2.2), by using the definition of the strong convexity of sets in Hilbert spaces (1), we obtain

$$\forall x \in \mathcal{M}, \ \forall p, q \in \mathrm{Exp}_x^{-1}(\mathcal{C}), \ \forall t \in [0, 1],$$

$$\text{if } z \in T_x\mathcal{M} : \|z - (tp + (1 - t)q)\| \leq \alpha t(1 - t)\|p - q\|^2, \ \text{then } z \in \mathrm{Exp}_x^{-1}(\mathcal{C}), \quad (14)$$

for some parameter $\alpha > 0$. We now consider arbitrary points $x, y \in \mathcal{M}$ and $\tilde{z} \in \mathcal{M}$ s.t. $d(\gamma(t), \tilde{z}) \leq \tilde{\alpha} t(1 - t)d^2(x, y)$, where $\gamma(t) : [0, 1] \to \mathcal{M}$ is the geodesic between $x$ and $y$ and $\tilde{\alpha} \overset{\text{def}}{=} \alpha \ell_\mathcal{M} L_\mathcal{M}^{-2} > 0$. Due to Assumption 1.3 we have

$$\ell_\mathcal{M} \|\mathrm{Exp}_{\gamma(t)}^{-1}(\tilde{z})\| \leq L_\mathcal{M}^2 \tilde{\alpha} t(1 - t)\|\mathrm{Exp}_x^{-1}(y)\|^2. \quad (15)$$

Now, (Martínez-Rubio and Pokutta, 2022, Corollary 24) using the Riemannian cosine law inequality for our Cartan-Hadamard manifold in the geodesic triangle with vertices $x$, $\tilde{z}$ and $\gamma(t)$, and the corresponding triangle in $T_x\mathcal{M}$ via $\mathrm{Exp}_x^{-1}(\cdot)$, we have

$$2\langle \mathrm{Exp}_x^{-1}(\tilde{z}), \mathrm{Exp}_x^{-1}(\gamma(t)) \rangle \geq \|\mathrm{Exp}_x^{-1}(\tilde{z})\|^2 + \|\mathrm{Exp}_x^{-1}(\gamma(t))\|^2 - \|\mathrm{Exp}_{\gamma(t)}^{-1}(\tilde{z})\|^2$$

$$2\langle \mathrm{Exp}_x^{-1}(\tilde{z}), \mathrm{Exp}_x^{-1}(\gamma(t)) \rangle = \|\mathrm{Exp}_x^{-1}(\tilde{z})\|^2 + \|\mathrm{Exp}_x^{-1}(\gamma(t))\|^2 - \|\mathrm{Exp}_x^{-1}(\tilde{z}) - \mathrm{Exp}_x^{-1}(\gamma(t))\|^2,$$

which implies

$$\|\mathrm{Exp}_x^{-1}(\tilde{z}) - \mathrm{Exp}_x^{-1}(\gamma(t))\|_x \leq \|\mathrm{Exp}_{\gamma(t)}^{-1}(\tilde{z})\|_{\gamma(t)}. \quad (16)$$

Now, let $p = \mathrm{Exp}_x^{-1}(y), q = \mathrm{Exp}_x^{-1}(x) = 0$, and $z = \mathrm{Exp}_x^{-1}(\tilde{z})$, and note that $\mathrm{Exp}_x^{-1}(\gamma(t)) = t\,\mathrm{Exp}_x^{-1}(y)$. Hence, combining (15) with (16) and using our new notation, we obtain

$$\ell_\mathcal{M} \|z - (tp + (1 - t)q)\|_x \leq L_\mathcal{M}^2 \tilde{\alpha} t(1 - t)\|p - q\|_x^2,$$

and after using the value of $\tilde{\alpha} = \alpha \ell_\mathcal{M} L_\mathcal{M}^{-2}$ and (14), we conclude that $z = \mathrm{Exp}_x^{-1}(\tilde{z}) \in \mathrm{Exp}_x^{-1}(\mathcal{C})$, and hence $\tilde{z} \in \mathcal{C}$. $\square$

**Proof of Proposition 2.7**

*Proof.* ($\Rightarrow$) We start with $\mathcal{C} \subset \mathcal{M}$ being an $\alpha$-geodesically strongly convex set. Now, let $z \in T_{\gamma(t)}\mathcal{M}$ such that

$$\|z\| \leq \frac{\alpha}{L_{\mathcal{M}}}t(1-t)d^2(x,y).$$

As the distance function $d$ can be bounded as $d(x,y) \leq L_{\mathcal{M}}\|\operatorname{Exp}_x^{-1}(y)\|$, we have

$$\|z\| \leq \frac{\alpha}{L_{\mathcal{M}}}t(1-t)d^2(x,y) \quad \Rightarrow \quad d\left(\gamma(t), \operatorname{Exp}_{\gamma(t)}(z)\right) \leq \alpha t(1-t)d^2(x,y)$$

As the set is geodesically strongly convex, we have $\operatorname{Exp}_{\gamma(t)}(z) \in \mathcal{C}$.

($\Leftarrow$) Now, we assume $\mathcal{C}$ to be a doubly exponentially strongly convex set with the parameter $\alpha$. We construct the point $z \in \mathcal{M}$ such that

$$d(\gamma(t), z) \leq \ell_{\mathcal{M}}\alpha t(1-t)d^2(x,y).$$

As the distance function $d$ can be bounded as $\ell_{\mathcal{M}}\|\operatorname{Exp}_{\gamma(t)}^{-1}(z)\| \leq d(\gamma(t), z)$, we have

$$d(\gamma(t), z) \leq \ell_{\mathcal{M}}\alpha t(1-t)d^2(x,y) \quad \Rightarrow \quad \|\operatorname{Exp}_{\gamma(t)}^{-1}(z)\| \leq \alpha t(1-t)d^2(x,y)$$

As the set is $\alpha$-double exponentially strongly convex, we have $z \in \mathcal{C}$. $\qquad\square$

**Proof of Proposition 3.3**

*Proof.* This proof is similar to Proposition B.1 until (31). We then write

$$u = \frac{1}{2}\gamma'_{x,v}(0) \text{ and } \omega = \left(\Gamma_x^{\gamma_{x,v}(1/2)}\right)^{-1}(\frac{\alpha}{4}d^2(x,v)z).$$

Hence, we have

$$\left\langle w; \operatorname{Exp}_x^{-1}(v)\right\rangle_x \geq \left\langle w; \operatorname{Exp}_x^{-1}\left(\operatorname{Exp}_x\left(u + \omega + R(u,v)\right)\right)\right\rangle_x.$$

Using the same arguments as those in the proof of Proposition B.1, we obtain

$$\left\langle w; \operatorname{Exp}_x^{-1}(v)\right\rangle_x \geq \frac{\alpha}{2}d^2(x,v)\|w\|_x + \left\langle w; R(\frac{1}{2}\gamma'_{x,v}(0), \Gamma_{\gamma_{x,v}(1/2)}^x(\frac{\alpha}{4}d^2(x,v)z^*)\right\rangle_x.$$

$\qquad\square$

**Proof of Theorem 4.2**

*Proof.* Let $s > 0$, and let us consider $(x,y) \in Q_s^2$ and write $\gamma$ as the geodesic between $x$ and $y$. On successively using the geodesic smoothness of $f$, Cauchy-Schwartz, and Lemma 4.1, for all smooth curves $c_t(\tilde{t}) : c_t(0) = \gamma(t)$ and $t, \tilde{t} \in [0,1]$, we obtain

$$
\begin{aligned}
f(c_t(\tilde{t})) - f^* &\leq f(\gamma(t)) - f^* + \langle\nabla f(\gamma(t)); \operatorname{Exp}_{\gamma(t)}^{-1}(c_t(\tilde{t}))\rangle_{\gamma(t)} + \frac{L}{2}d^2(\gamma(t), c_t(\tilde{t})) \\
&\leq f(\gamma(t)) - f^* + \|\nabla f(\gamma(t))\|_{\gamma(t)}\|\operatorname{Exp}_{\gamma(t)}^{-1}(c_t(\tilde{t}))\|_{\gamma(t)} + \frac{L}{2}d^2(\gamma(t), c_t(\tilde{t})) \\
&\leq f(\gamma(t)) - f^* + \sqrt{2L(f(\gamma(t)) - f^*)}\ell_{\mathcal{M}}^{-1}d(\gamma(t), c_t(\tilde{t})) + \frac{L}{2}d^2(\gamma(t), c_t(\tilde{t})) \\
&\leq \left(\sqrt{f(\gamma(t)) - f^*} + \sqrt{\frac{L\max\{\ell_{\mathcal{M}}^{-2}; 1\}}{2}}d(\gamma(t), c_t(\tilde{t}))\right)^2.
\end{aligned}
\tag{17}
$$

Therefore, to ensure that $c_t(\tilde{t}) \in Q_s$, we can identify a sufficient condition on $d(\gamma(t), c_t(\tilde{t}))$, such that

$$\left( \sqrt{f(\gamma(t)) - f^*} + \sqrt{\frac{L \max\{\ell_{\mathcal{M}}^{-2}; 1\}}{2}} d(\gamma(t), c_t(\tilde{t})) \right)^2 \leq s,$$

$$\Leftarrow d(\gamma(t), c_t(\tilde{t})) \leq \text{sufficient condition} \leq \sqrt{\frac{2}{L \max\{\ell_{\mathcal{M}}^{-2}; 1\}}} \left( \sqrt{s} - \sqrt{f(\gamma(t)) - f^*} \right) \quad (18)$$

As $f$ is strongly convex, on using Definition C.18 and because $(x, y) \in Q_s$, we obtain

$$f(\gamma(t)) - f^* \leq s - (1 - t)t\frac{\mu}{2}d^2(x, y),$$

Since $\sqrt{\cdot}$ is a concave function, we have $\sqrt{x - y} \leq \sqrt{x} - \frac{y}{2\sqrt{x}}$. Therefore,

$$\sqrt{f(\gamma(t)) - f^*} \leq \sqrt{s - (1 - t)t\frac{\mu d^2(x, y)}{2}} \leq \sqrt{s} - \frac{(1 - t)t\mu d^2(x, y)}{4\sqrt{s}}.$$

Hence, we have

$$\sqrt{\frac{2}{L \max\{\ell_{\mathcal{M}}^{-2}; 1\}}} \left( \sqrt{s} - \sqrt{f(\gamma(t)) - f^*} \right) \geq \sqrt{\frac{2}{L \max\{\ell_{\mathcal{M}}^{-2}; 1\}}} \left( \frac{(1 - t)t\mu d^2(x, y)}{4\sqrt{s}} \right) \quad (19)$$

$$= (1 - t)t\frac{\mu}{2\sqrt{2sL \max\{\ell_{\mathcal{M}}^{-2}; 1\}}}d^2(x, y). \quad (20)$$

Therefore,

$$d(\gamma(t), c_t(\tilde{t})) \leq (1 - t)t\frac{\mu}{2\sqrt{2sL \max\{\ell_{\mathcal{M}}^{-2}; 1\}}}d^2(x, y)$$

$$\overset{(20)}{\Longrightarrow} d(\gamma(t), c_t(\tilde{t})) \leq \sqrt{\frac{2}{L \max\{\ell_{\mathcal{M}}^{-2}; 1\}}} \left( \sqrt{s} - \sqrt{f(\gamma(t)) - f^*} \right)$$

$$\overset{(18)}{\Longrightarrow} \left( \sqrt{f(\gamma(t)) - f^*} + \sqrt{\frac{L \max\{\ell_{\mathcal{M}}^{-2}; 1\}}{2}} d(\gamma(t), c_t(\tilde{t})) \right)^2 \leq s$$

$$\overset{(17)}{\Longrightarrow} f(c_t(\tilde{t})) - f^* \leq s.$$

Hence, $c_t(\tilde{t})$ is in the set $Q_s$, which is the definition of geodesic strong convexity. $\qquad\square$

**Proof of Theorem 5.1**

*Proof.* This proof is based on (Demyanov and Rubinov, 1970; Garber and Hazan, 2015; Kerdreux et al., 2021b), but in in a Riemannian setting. Using the geodesic smoothness (Definition C.19) of $f$ at $x_{t+1} = \gamma(s_t)$, we obtain

$$f(x_{t+1}) \leq f(x_t) - \left\langle -\nabla f(x_t); \text{Exp}_{x_t}^{-1}(\gamma(s_t)) \right\rangle_x + \frac{L}{2}d^2(x_t, \gamma(s_t)).$$

As $\gamma$ is a geodesic between $x_t$ and $v_t$, we have $d(x_t, \gamma(s_t)) = s_t d(x_t, v_t)$ and $\text{Exp}_{x_t}^{-1}(\gamma(s_t)) = s_t \text{Exp}_{x_t}^{-1}(v_t)$. Hence, we now have

$$f(x_{t+1}) \leq f(x_t) - s_t\left\langle -\nabla f(x_t); \text{Exp}_{x_t}^{-1}(v_t) \right\rangle_x + \frac{L}{2}s_t^2 d^2(x_t, v_t).$$

According to the short-step rule for $s_t$ (Algorithm 1), for all $s \in [0, 1]$, we now have

$$f(x_{t+1}) \leq f(x_t) - s\left\langle -\nabla f(x_t); \text{Exp}_{x_t}^{-1}(v_t) \right\rangle_x + \frac{L}{2}s^2 d^2(x_t, v_t).$$

After using the optimality of $v_t$, we have $\langle -\nabla f(x_t); \mathrm{Exp}_{x_t}^{-1}(v_t)\rangle_x \leq \langle -\nabla f(x_t); \mathrm{Exp}_{x_t}^{-1}(x^*)\rangle_x$, where $x^* \in \mathcal{C}$ is a solution to (OPT). Then, owing to the geodesic convexity of $f$, we have

$$f(x^*) - f(x) \geq \langle \nabla f(x); \mathrm{Exp}_x^{-1}(x^*)\rangle_x.$$

Hence, as it is the case in the Hilbertian setting, the FW gap $\langle -\nabla f(x_t); \mathrm{Exp}_{x_t}^{-1}(v_t)\rangle_x$ upper bounds the primal gap at $x_t$, i.e.,

$$f(x_t) - f(x^*) \leq \langle -\nabla f(x_t); \mathrm{Exp}_{x_t}^{-1}(v_t)\rangle_x.$$

We write $h_t = f(x_t) - f(x^*)$, and we hence have

$$h_{t+1} \leq h_t(1 - s/2) - s/2\langle -\nabla f(x_t); \mathrm{Exp}_{x_t}^{-1}(v_t)\rangle_x + \frac{L}{2}s^2 d^2(x_t, v_t). \tag{21}$$

Now, with $c = \mathrm{argmin}_{x\in\mathcal{C}} \|\nabla f(x)\|_x > 0$, after using the (RSI) at $x_t$ and since $-\nabla f(x_t) \in T_{x_t}\mathcal{M}$, we obtain

$$\langle -\nabla f(x_t); \mathrm{Exp}_{x_t}^{-1}(v_t)\rangle_x \geq \alpha c d(x_t, v_t)^2,$$

such that, for all $s \in [0,1]$, we have

$$h_{t+1} \leq h_t(1 - s/2) + \frac{s}{2}\Big(Ls - \alpha c\Big)d^2(x_t, v_t). \tag{22}$$

Then, if $\alpha c/L < 1$, by choosing $s = \alpha c/L$ in (22), we have $h_{t+1} \leq h_t(1 - \alpha c/(2L))$; else, we have $L - \alpha c < 0$, and on selecting $s = 1$, we simply have $h_{t+1} \leq h_t/2$. Hence,

$$h_{t+1} \leq h_t \max\{1/2, 1 - \alpha c/(2L)\}.$$

$\square$

**Proof of Theorem 5.2**

*Proof.* As in the proof of Theorem (5.1), (21) is satisfied, i.e.,

$$h_{t+1} \leq h_t(1 - s_t/2) - s_t/2\langle -\nabla f(x_t); \mathrm{Exp}_{x_t}^{-1}(v_t)\rangle_{x_t} + \frac{L}{2}s_t^2 d^2(x_t, v_t). \tag{23}$$

Now, from Proposition 3.3, as $\mathcal{C}$ is double geodesically $\alpha$-strongly convex, an approximate RSI is satisfied (Definition 3.2) at $x_t$ with $-\nabla f(x_t) \in T_{x_t}\mathcal{M}$ with the residual $r(x_t)$ as in (Residual), i.e.,

$$r(x_t) = R_{x_t}\Big[\frac{1}{2}\gamma'_{x_t,v_t}(0), \Gamma_{\gamma_{x_t,v_t}(1/2)}^{x_t}\Big(\frac{\alpha}{4}d^2(x_t, v_t)z^*\Big)\Big]. \tag{24}$$

Hence, on combining with $\mathrm{argmin}_{x\in\mathcal{C}} \|\nabla f(x)\|_x > c$, we can lower-bound $\langle -\nabla f(x_t); \mathrm{Exp}_{x_t}^{-1}(v_t)\rangle_{x_t}$ as follows:

$$\langle -\nabla f(x_t); \mathrm{Exp}_{x_t}^{-1}(v_t)\rangle_{x_t} \geq \alpha c d(x_t, v_t)^2 + \langle -\nabla f(x); r(x_t)\rangle_{x_t}.$$

On substituting this inequality in (23), for all $s \in [0,1]$, we have

$$h_{t+1} \leq h_t(1 - s/2) + \frac{s}{2}\Big(Ls - \alpha c\Big)d^2(x_t, v_t) + \frac{s}{2}\langle \nabla f(x_t); r(x_t)\rangle_{x_t}. \tag{25}$$

We use (12) to upper-bound the term $\|r(x_t)\|_{x_t}$. Hence, we are first required to obtain an upper bound on $\|\gamma'_{x_t,v_t}(0)\|_{x_t}$ and $\|\frac{\alpha}{4}d^2(x_t, v_t)z^*\|_{\gamma_{x_t,v_t}(1/2)}$. We first note that

$$\Big\|\frac{\alpha}{4}d^2(x_t, v_t)z^*\Big\|_{\gamma_{x_t,v_t}(1/2)} = \frac{\alpha}{4}d^2(x_t, v_t).$$

Furthermore, by definition of the exponential mapping, we have $\|\gamma'_{x_t,v_t}(0)\|_{x_t} = \|\mathrm{Exp}_{x_t}^{-1}(v_t)\|_x$, and according to Assumption 1.3, we have

$$\|\gamma'_{x_t,v_t}(0)\|_{x_t} \leq \frac{1}{\ell_{\mathcal{M}}}d(x_t, v_t).$$

When plugging these two bounds in the residual $r(x_t)$ (24), with the growth condition on the residual (12), we have

$$
\|r(x_t)\|_{x_t} \quad \leq \quad C \, \max \Big\{ \frac{\alpha^2}{16\ell_{\mathcal{M}}} d^5(x_t, v_t); \frac{\alpha}{4\ell_{\mathcal{M}}^2} d^4(x_t, v_t) \Big\} \tag{26}
$$

$$
\|r(x_t)\|_{x_t} \quad \leq \quad C d^4(x_t, v_t) \, \max \Big\{ \frac{\alpha^2}{16\ell_{\mathcal{M}}} d(x_t, v_t); \frac{\alpha}{4\ell_{\mathcal{M}}^2} \Big\}. \tag{27}
$$

On using $d(x_t, v_t) \leq \delta$, and with $\tilde{C} := C \, \max \big\{ \alpha^2/(16\ell_{\mathcal{M}})\delta; \alpha/(4\ell_{\mathcal{M}}^2) \big\}$, we have $\|r(x_t)\|_{x_t} \leq \tilde{C} d^4(x_t, v_t)$. Hence, using the Cauchy-Schwartz inequality and $\|\nabla f(x_t)\|_x \leq \delta L$, (25) becomes

$$
h_{t+1} \quad \leq \quad h_t(1 - s/2) + \frac{s}{2}\Big( Ls - \alpha c \Big) d^2(x_t, v_t) + \frac{s}{2}\delta L\tilde{C} d^4(x_t, v_t) \tag{28}
$$

$$
\leq \quad h_t(1 - s/2) + \frac{s}{2} d^2(x_t, v_t)\Big( Ls - \alpha c + \delta L\tilde{C} d^2(x_t, v_t) \Big). \tag{29}
$$

As we assumed $d(x_t, v_t)^2 \leq (\alpha c)/(2\delta L\tilde{C})$, we have $-\alpha c + \delta L\tilde{C} d^2(x_t, v_t) \leq -(\alpha c)/2 < 0$. Let us consider $s^* := (\alpha c - \delta L\tilde{C} d^2(x_t, v_t))/L > 0$. If $s^* > 1$, then the choice of $s = 1$ results in $h_{t+1} \leq h_t/2$. Else, $s^* \in [0, 1]$, and we select $s = s^*$. We hence obtain $h_{t+1} \leq h_t(1 - s^*/2) \leq 1 - (\alpha c)/(2L)$. Overall, we obtain

$$
h_{t+1} \leq h_t \max\{1/2, 1 - \alpha c/(2L)\}.
$$

$\square$

# B TECHNICAL PROPOSITION AND LEMMAS

## B.1 DOUBLE GEODESIC STRONG CONVEXITY IMPLIES RSI

**Proposition B.1** (Double Geodesic Str. Cvx. implies RSI). *Let $\mathcal{C} \subset \mathcal{M}$ be a double geodesic $\alpha$-strongly convex set (Definition 2.3) in a complete connected Riemannian manifold $\mathcal{M}$. Let us assume that the double exponential map operator (Definition 3.1) is such that*

$$h_x(u, v) = u + v, \quad \forall (u, v) \in T_x\mathcal{M}. \tag{30}$$

*The RSI in Definition 2.4 is then satisfied.*

*Proof.* Let us consider $x \in \mathcal{C}$, $w \in T_x\mathcal{M}$, and $v \in \mathcal{C}$ s.t.

$$v \in \operatorname*{argmax}_{z \in \mathcal{C}} \langle w; \operatorname{Exp}_x^{-1}(z) \rangle_x.$$

Then, by (8), which is equivalent to Definition 2.3 with $t = 1/2$, we have $\operatorname{Exp}_{\gamma_{x,v}(1/2)} \left( \frac{\alpha}{4} d^2(x, v) z \right) \in \mathcal{C}$, where $\gamma_{x,v}(\cdot)$ is the geodesic joining $x$, and $v$ and $z$ are a unit norm vector in $T_{\gamma_{x,v}(1/2)}\mathcal{M}$. Then, by optimality of $v$, for all $z \in T_{\gamma_{x,v}(1/2)}\mathcal{M}$ with $\|z\|_{\gamma_{x,v}(1/2)} = 1$, we have

$$\left\langle w; \operatorname{Exp}_x^{-1}(v) \right\rangle_x \geq \left\langle w; \operatorname{Exp}_x^{-1} \left( \operatorname{Exp}_{\gamma_{x,v}(1/2)} \left( \frac{\alpha}{4} d^2(x, v) z \right) \right) \right\rangle_x.$$

Let us first recall that, by definition of the exponential map, for the geodesic $\gamma_{x,v}$ and for all $t \in [0, 1]$, we have $\operatorname{Exp}_x(t\gamma'_{x,v}(0)) = \gamma_{x,v}(t)$ such that we can write

$$\left\langle w; \operatorname{Exp}_x^{-1}(v) \right\rangle_x \geq \left\langle w; \operatorname{Exp}_x^{-1} \left( \operatorname{Exp}_{\operatorname{Exp}_x(\frac{1}{2}\gamma'_{x,v}(0))} \left( \frac{\alpha}{4} d^2(x, v) z \right) \right) \right\rangle_x. \tag{31}$$

We can hence write this in terms of the double exponential map and subsequently in terms of the exponential operator map. We use

$$\operatorname{Exp}_{\operatorname{Exp}_x(\frac{1}{2}\gamma'_{x,v}(0))} \left( \frac{\alpha}{4} d^2(x, v) z \right) = \operatorname{Exp}_x \left( \frac{1}{2}\gamma'_{x,v}(0); \left( \Gamma_x^{\gamma_{x,v}(1/2)} \right)^{-1} \left( \frac{\alpha}{4} d^2(x, v) z \right) \right).$$

Hence, on using the assumption (30) on the exponential operator, we have

$$\operatorname{Exp}_{\operatorname{Exp}_x(\frac{1}{2}\gamma'_{x,v}(0))} \left( \frac{\alpha}{4} d^2(x, v) z \right) = \operatorname{Exp}_x \left( \frac{1}{2}\gamma'_{x,v}(0) + \left( \Gamma_x^{\gamma_{x,v}(1/2)} \right)^{-1} \left( \frac{\alpha}{4} d^2(x, v) z \right) \right).$$

When plugging the last equality in (31), we obtain

$$\left\langle w; \operatorname{Exp}_x^{-1}(v) \right\rangle_x \geq \frac{1}{2} \left\langle w; \gamma'_{x,v}(0) \right\rangle_x + \left\langle w; \left( \Gamma_x^{\gamma_{x,v}(1/2)} \right)^{-1} \left( \frac{\alpha}{4} d^2(x, v) z \right) \right\rangle_x.$$

It should be noted that $\gamma'_{x,v}(0) = \operatorname{Exp}_x^{-1}(v)$, such that

$$\left\langle w; \operatorname{Exp}_x^{-1}(v) \right\rangle_x \geq 2 \left\langle w; \left( \Gamma_x^{\gamma_{x,v}(1/2)} \right)^{-1} \left( \frac{\alpha}{4} d^2(x, v) z \right) \right\rangle_x.$$

With $\left( \Gamma_x^y \right)^{-1} = \Gamma_y^x$ and the isometry property of $\Gamma_x^{\gamma_{x,v}(1/2)}$, we have

$$\left\langle w; \Gamma_{\gamma_{x,v}(1/2)}^x \left( \frac{\alpha}{4} d^2(x, v) z \right) \right\rangle_x = \left\langle \Gamma_{\gamma_{x,v}(1/2)}^x \Gamma_x^{\gamma_{x,v}(1/2)} w; \Gamma_{\gamma_{x,v}(1/2)}^x \left( \frac{\alpha}{4} d^2(x, v) z \right) \right\rangle_x \tag{32}$$

$$= \left\langle \Gamma_x^{\gamma_{x,v}(1/2)} w; \frac{\alpha}{4} d^2(x, v) z \right\rangle_{\gamma_{x,v}(1/2)}. \tag{33}$$

Hence, for all $z$ of a unit norm in $T_{\gamma_{x,v}(1/2)}\mathcal{M}$, we obtain

$$\left\langle w; \operatorname{Exp}_x^{-1}(v) \right\rangle_x \geq 2 \left\langle \Gamma_x^{\gamma_{x,v}(1/2)} w; \frac{\alpha}{4} d^2(x, v) z \right\rangle_{\gamma_{x,v}(1/2)} \tag{34}$$

$$= \frac{\alpha}{2} d^2(x, v) \left\langle \Gamma_x^{\gamma_{x,v}(1/2)} w; z \right\rangle_{\gamma_{x,v}(1/2)}. \tag{35}$$

Furthermore, by maximizing over $z$, for the best $z^*$, we obtain

$$\left\langle \Gamma_x^{\gamma_{x,v}(1/2)} w; z^* \right\rangle_{\gamma_{x,v}(1/2)} = \|\Gamma_x^{\gamma_{x,v}(1/2)} w\|_{\gamma_{x,v}(1/2)}.$$

Then, because the parallel transport $\Gamma_x^{\gamma_{x,v}(1/2)}$ is an isometry, we finally have

$$\left\langle w; \operatorname{Exp}_x^{-1}(v) \right\rangle_x \geq \frac{\alpha}{2} d^2(x, v) \|\Gamma_x^{\gamma_{x,v}(1/2)} \nabla f(x)\|_{\gamma_{x,v}(1/2)} = \frac{\alpha}{2} d^2(x, v) \|w\|_x.$$

$\square$

## B.2 Proof of Smoothness Property Lemma

First, we introduce the Fenchel conjugate of a function defined on a manifold.

**Definition B.2.** *(Bergmann et al., 2021) Let us Suppose that $f : \mathcal{C} \to \mathbb{R}$, where $\mathcal{C} \subset \mathcal{M}$ is a strictly convex set, where $\mathcal{M}$ is a Cartan-Hadamard manifold. For $m \in \mathcal{M}$, the $m$-Fenchel conjugate of $f$ is defined as the function $f_m^* : T_m^*\mathcal{M} \to \mathbb{R}$ such that*

$$f_m^*(\xi_m) := \sup_{x \in T_m\mathcal{M}} \left\{ \langle \xi_m, x \rangle - f(\operatorname{Exp}_m x) \right\}, \tag{36}$$

*where $T_m^*\mathcal{M}$ is the cotangent bundle of $T_m\mathcal{M}$*

In particular, we need the following property.

**Lemma B.3.** *(Bergmann et al., 2021, lem. 3.7) Let us suppose that $f, \tilde{f} : \mathcal{C} \to \mathbb{R}$ are proper functions, where $\mathcal{C} \subset \mathcal{M}$ is a strictly convex set, and $\mathcal{M}$ is a Cartan-Hadamard manifold, and let $m \in \mathcal{C}$. Then,*

$$\text{if } f(p) \leq \tilde{f}(p) \ \forall p \in \mathcal{C}, \text{ then } f_m^*(\xi_m) \geq \tilde{f}_m^*(\xi_m) \ \forall \xi_m \in T_m^*\mathcal{M}. \tag{37}$$

We are now ready to prove Lemma 4.1.

**Lemma 4.1** (Smoothness property). *Let $f$ be a geodesically $L$-smooth function (Definitions C.18 and C.19) defined on the geodesically closed subset $\mathcal{C} \subset \mathcal{M}$, where $\mathcal{M}$ is an Cartan-Hadamard manifold. We denote $x^* \in \operatorname{argmin}_{x \in \mathcal{C}} f(x)$. Then,*

$$\|\nabla f(x)\|_x \leq \sqrt{2L(f(x) - f(x^*))}. \tag{11}$$

*Proof.* As $f$ is a geodesically smooth function, for all $p \in \mathcal{C}$, we have $f(p) \leq \tilde{f}(p)$, where

$$\tilde{f}(p) = f(x) + \langle \nabla f(x), \operatorname{Exp}_x^{-1}(p) \rangle + \frac{L}{2}d^2(x, p), \quad \forall p \in \mathcal{C}.$$

Therefore, according to Lemma B.3, we have $f_m^*(\xi_m) \geq \tilde{f}_m^*(\xi_m)$, for all $m \in \mathcal{M}$. Using the definition (36),

$$\tilde{f}_m^*(\xi_m) = \sup_{Z \in T_m\mathcal{M}} \langle \xi_m, Z \rangle_x - \tilde{f}(\operatorname{Exp}_m(Z))$$

In the particular case wherein $m = x$, we obtain

$$\tilde{f}_x^*(\xi_x) = \sup_{Z \in T_m\mathcal{M}} \langle \xi_x, Z \rangle - f(x) - \langle \nabla f(x), Z \rangle_x - \frac{L}{2}\|Z\|_x^2$$

$$= -f(x) + \frac{1}{2L}\|\xi_x - \nabla f(x)\|_x^2$$

Therefore, the inequality $f_x^*(\xi_x) \geq \tilde{f}_x^*(\xi_x)$ can be written as

$$\frac{1}{2L}\|\xi_x - \nabla f(x)\|_x^2 \leq f_x^*(\xi_x) + f(x).$$

In particular, at $\xi_x = 0$, we have the desired result, as

$$f_x^*(0) = \sup_{Z \in T_m\mathcal{M}} \langle 0, Z \rangle_x - f(\operatorname{Exp}_x(Z)) = \sup_{Z \in T_m\mathcal{M}} -f(\operatorname{Exp}_x(Z)) = -\inf_{z \in \mathcal{C}} f(z) = -f^*.$$

$\square$

## C    Preliminaries and Notations

Our primary goal is defining several notions of strong convexity for sets in metric spaces that are well-suited for analyzing and designing optimization algorithms. In this study, we first focus on the case of complete and connected $n$-dimensional Riemannian manifolds. We now recall some useful concepts.

### C.1    Geodesic Metric Spaces

**Definition C.1** (Metric space). *A metric space is a pair $(\mathcal{M}, d_{\mathcal{M}})$, where $\mathcal{M}$ is a set, and $d_{\mathcal{M}} : \mathcal{M} \times \mathcal{M} \to \mathbb{R}^+$ is a distance function (the metric), i.e., it satisfies positivity, symmetry, and the triangle inequality.*

**Definition C.2** (Geodesics). *A geodesic $\gamma(t) : [0, 1] \to \mathcal{M}$ between two points $x, y \in \mathcal{M}$ is a smooth curve such that $\gamma(0) = x$, $\gamma(1) = y$, and $\gamma''(t) = 0$.*

**Definition C.3** (Geodesic metric space). *A metric set is a geodesic metric space if, for all $x, y \in \mathcal{M}$, there exists a geodesic that connects $x$ and $y$. When this geodesic is unique, the set is considered uniquely geodesic.*

**Definition C.4** (Non-positively curved spaces). *If, for all geodesics $\gamma$ and for all elements $x \in \mathcal{M}$, the metric $d_{\mathcal{M}}$ satisfies (e.g., (Paris, 2021, Corollary 2.5.))*

$$d_{\mathcal{M}}(x, \gamma(t))^2 \leq (1 - t)d_{\mathcal{M}}(x, \gamma(0))^2 + td_{\mathcal{M}}(x, \gamma(1))^2 - t(1 - t)d_{\mathcal{M}}(\gamma(0), \gamma(1))^2, \qquad (38)$$

*then $\mathcal{M}$ is a non-positively curved space.*

This non-positive curvature indicates that the function $f(\cdot) = d_{\mathcal{M}}(\cdot, x)^2$ is geodesically strongly convex (see Definition C.18) for all reference point $x \in \mathcal{M}$. In particular, *Cartan–Hadamard spaces* are geodesic metric spaces generalizing Hilbert spaces which are well-suited for convex optimization purposes (Jost, 2012; Bridson and Haefliger, 2013; Bacák, 2014; Bacak, 2018).

**Definition C.5** (Cartan–Hadamard spaces). *A Cartan–Hadamard space $\mathcal{M}$ (a.k.a. CAT(0) space or space of non-positive curvature) is a geodesically metric space that is non-positively curved.*

### C.2    Riemannian Manifolds

In this section, we assume that $\mathcal{M}$ is a $n$-dimensional Riemannian manifold, for instance, the group of rotations $\mathrm{SO}(n)$; hyperbolic spaces; spheres; or the manifold of symmetric positive definite matrices.

**Definition C.6** (Connected manifolds). *A manifold $\mathcal{M}$ is connected if, for all $x, y \in \mathcal{M}$, there exists a continuous path joining those points.*

**Definition C.7** (Tangent space). *For a point $x \in \mathcal{M}$, a tangent vector is the tangent of a parameterized curve passing through $x$, and all the tangent vectors at $x$ form the tangent space $T_x\mathcal{M}$.*

**Definition C.8** (Riemannian Metric (Metric Tensor)). *Let $\mathcal{M}$ be a Riemannian manifold. A Riemannian metric on $\mathcal{M}$ is a smooth (0,2)-tensor field*

$$g : x \in \mathcal{M} \mapsto g_x \in \mathrm{Bil}(T_x\mathcal{M} \times T_x\mathcal{M}, \mathbb{R})$$

*such that for each point $x \in \mathcal{M}$, the bilinear form $g_x$ satisfies symmetry, positive-definiteness and smoothness. The metric tensor $g$ equips $\mathcal{M}$ with a notion of angle, length, and volume.*

**Definition C.9** (Inner Product at a Point). *Given a Riemannian manifold $(\mathcal{M}, g)$, the inner product at a point $x \in \mathcal{M}$ is the bilinear form*

$$\langle \cdot, \cdot \rangle_x := g_x : T_x\mathcal{M} \times T_x\mathcal{M} \to \mathbb{R}$$

*which is symmetric and positive-definite.*

**Definition C.10** (Complete Manifold). *A Riemannian manifold $(\mathcal{M}, g)$ is complete if, for all $x \in \mathcal{M}$, the exponential map $Exp_x$ is defined in the whole tangent space $T_x\mathcal{M}$.*

**Complete Riemannian manifold and Riemannian metric**    A connected Riemannian manifold is also a metric space. Furthermore, the length of a continuous piecewise smooth path $\gamma : [0, 1] \to \mathcal{M}$ is defined as $L(\gamma) := \int_0^1 \|\gamma'(t)\|_{\gamma(t)} dt$. We denote $\mathcal{P}_{x,y}$ as the set of continuous piecewise smooth paths joining $x$ and $y$. We have the *Riemannian metric* $d_{\mathcal{M}}(x, y) := \inf_{\gamma \in \mathcal{P}_{x,y}} L(\gamma)$ as a distance (Boumal, 2020, Theorem 10.2) and $(\mathcal{M}, d_{\mathcal{M}})$ is a metric space.

## C.3 Sets in Riemannian Manifolds

**Importance of sets when optimizing on Manifolds.** When $\mathcal{M}$ is a connected compact Riemannian manifold, and $f$ is a geodesically convex function on $\mathcal{M}$ (Definition C.18), then $f$ is constant (Boumal, 2020, Corollary 11.10). This result motivates the optimization of geodesically convex functions over subsets $\mathcal{C}$ of these manifolds.

**Assumptions for Sets in Riemannian Manifolds** In this paper, we will make the assumption that we are working on a subset $\mathcal{C} \subset \mathcal{M}$ that is *uniquely geodesically convex*, defined as the set $\mathcal{C}$ being both convex and uniquely geodesic.

**Definition C.11** (Geodesic Convex Closed Subset of a Manifold)**.** *We say that a closed subset $\mathcal{C}$ of a manifold $\mathcal{M}$ is convex if for any two points $x, y \in \mathcal{C}$, there exists a geodesic from $x$ to $y$ that is distance minimizing and is contained in $\mathcal{C}$.*

**Definition C.12** (Uniquely Geodesic Subset of a Manifold)**.** *We say that a subset $\mathcal{C}$ of a manifold $\mathcal{M}$ is uniquely geodesic if for every two points $x, y \in \mathcal{C}$, there is only one geodesic between $x$ and $y$ that is contained in $\mathcal{C}$.*

**Assumption C.13.** *The set $\mathcal{C} \subseteq \mathcal{M}$ is compact, convex, and uniquely geodesic.*

For instance, open hemispheres and Cartan–Hadamard manifolds, that is, complete simply connected manifolds of non-positive sectional curvature everywhere, are uniquely geodesic. Hence, their compact counterparts are geodesically convex subsets. However, a closed hemisphere is not uniquely geodesic.

The *exponential map* parameterizes the manifold by mapping vectors in the tangent spaces to $\mathcal{M}$. For $x \in \mathcal{M}$, it is in general a local diffeomorphism; however, when $\mathcal{M}$ is complete (Hopf and Rinow, 1931), it is defined on the whole tangent space as follows.

**Definition C.14** (Exponential Map)**.** *Let $\mathcal{M}$ be a Riemannian manifold. For all $x \in \mathcal{M}$ and $v \in T_x\mathcal{M}$, let the geodesic $\gamma : [0,1] \to \mathcal{M}$ satisfying $\gamma(0) = x$ with $\gamma'(0) = v$. The exponential map at $x$, $\mathrm{Exp}_x : T_x\mathcal{M} \to \mathcal{M}$, is defined as $\mathrm{Exp}_x(v) := \gamma(1)$.*

**Remark 1** (Bijective Exponential Map and Logarithmic Map in the Right Domain)**.** *For uniquely geodesically convex sets $\mathcal{C}$ and $x \in \mathcal{C}$, define the set*

$$\mathcal{C}_x \stackrel{\text{def}}{=} \{v \in T_x\mathcal{M} : t \mapsto \mathrm{Exp}_x(tv), \text{ for } t \in [0,1], \text{ is minimizing and } \mathrm{Exp}_x(v) \in \mathcal{C}\},$$

*then, the exponential map from $x$ restricted to this set is well defined and bijective in $C$, that is, $\mathrm{Exp}_x(y) : \mathcal{C}_x \to \mathcal{C}$ is a bijective function. In this work we will always refer to the inverse of the exponential map, or logarithmic map, as the inverse of this restriction: $\mathrm{Exp}_x^{-1} : \mathcal{C} \to \mathcal{C}_x$. Note that generally in the literature the logarithmic map is not defined using this restriction.*

The exponential map links the manifold and its tangent spaces, but vectors belonging to different tangent spaces are not directly comparable. Using the Levi-Civita connection, we can define a parallel transport operator between two points $x, y \in \mathcal{M}$ in the manifold (Ambrose, 1956).

**Definition C.15** (Parallel Transport Operator)**.** *For all $x, y \in \mathcal{M}$, the parallel transport map $\Gamma_x^y : T_x\mathcal{M} \to T_y\mathcal{M}$ combines vectors from different tangent spaces by transporting them along geodesics and such that $\langle \Gamma_x^y u; \Gamma_x^y v \rangle_y = \langle u; v \rangle_x$ for all $(u, v) \in T_x\mathcal{M}$.*

## C.4 Functions over Manifolds

We first define the Riemannian gradient of $f$ at $x$ as the unique element $\nabla f(x) \in T_x\mathcal{M}$, such that the directional derivative $Df(x)[v] = \langle v; \nabla f(x) \rangle_x$ for all $v \in T_x\mathcal{M}$.

We now recall the notion of relative geodesic (strong) convexity and smoothness for a function $f : \mathcal{M} \to \mathbb{R}$ w.r.t. a distance function $d : \mathcal{M} \times \mathcal{M} \to \mathbb{R}^+$ (see, for instance, (Zhang and Sra, 2016)).

**Definition C.16** (Distance function)**.** *A function $d : \mathcal{M} \times \mathcal{M} \to \mathbb{R}^+$ is called a distance function if it satisfies positivity, symmetry, and the triangle inequality.*

**Remark 2.** *The distance $d$ could differ from the distance function $d_\mathcal{M}$ associated with the manifold $\mathcal{M}$.*

**Definition C.17** (Euclidean Smoothness and Strong Convexity). *In the euclidean case, a function is called L-smooth, μ-strongly convex if the function is differentiable and satisfies, $\forall x, y \in dom f$*

$$\frac{\mu}{2}\|y - x\|^2 \leq f(y) - f(x) - \nabla f(x)(y - x) \leq \frac{L}{2}\|y - x\|^2$$

In the case of geodesic space, this definition takes the following forms:

**Definition C.18** (Geodesic (Strong) Convexity). *Let $\mathcal{C}$ be a uniquely geodesic set. A function $f : \mathcal{C} \subseteq \mathcal{M} \to \mathbb{R}$ is geodesically μ-strongly convex in $\mathcal{C}$ (resp. convex if $\mu = 0$) w.r.t. $d$ if, for all $x, y \in \mathcal{C}$, we have*

$$\forall t \in [0, 1], \quad f(\gamma(t)) \leq (1 - t)f(x) + tf(y) - \frac{\mu}{2}t(1 - t)d^2(x, y), \quad \mu \geq 0. \tag{39}$$

**Definition C.19** (Geodesic Smoothness). *Let $\mathcal{C}$ be a uniquely geodesic set. A function $f : \mathcal{C} \subseteq \mathcal{M} \to \mathbb{R}$ is geodesically L-smooth in $\mathcal{C}$ w.r.t. $d$ if, for all $x, y \in \mathcal{C}$, we have*

$$|f(y) - f(x) - \langle \nabla f(x); \operatorname{Exp}_x^{-1}(y)\rangle_x| \leq \frac{L}{2}d^2(x, y), \quad L \geq 0.$$

Note that when the function is also geodesic (strongly) convex, the smoothness condition can be simplified by removing the absolute value. Indeed, (strong) convexity implies that the argument of the absolute value is nonnegative.

Finally, we state the following assumption on the relation between the distance function used to define smoothness and strong convexity and the distance function associated with the manifold $\mathcal{M}$. This assumption allows us to use distances whose value is within some constant factors from the Riemannian distance $\|\operatorname{Exp}_x^{-1}(y)\|$, which corresponds to the definition below with $\ell_{\mathcal{M}} = L_{\mathcal{M}} = 1$.

**Assumption C.20** (Distances Equivalence). *Let $\mathcal{C} \subset \mathcal{M}$ satisfies Assumption 1.2 and let $d : \mathcal{M} \times \mathcal{M} \to \mathbb{R}$ be a distance function, possibly different from the Riemannian distance. There exists $0 < l_{\mathcal{M}} \leq L_{\mathcal{M}}$ such that for all pair $(x, y) \in \mathcal{C}^2$,*

$$\ell_{\mathcal{M}}\|\operatorname{Exp}_x^{-1}(y)\|_x \leq d(x, y) \leq L_{\mathcal{M}}\|\operatorname{Exp}_x^{-1}(y)\|_x. \tag{40}$$

# D EXAMPLES OF RIEMANNIAN STRONGLY CONVEX SETS: BALLS WITH RESTRICTED RADIUS

In this section, we always make use of the Riemannian distance $d_{\mathcal{M}}(x, y) = \| \operatorname{Exp}_x^{-1}(y)\|$. The Riemannian strongly convex set, in Eq. (7), is the most restrictive of our definitions. It implies that, for all points $x$ in the set $\mathcal{C}$, the logarithmic image of $\mathcal{C}$ around $x$, i.e., $\operatorname{Exp}_x^{-1}(\mathcal{C})$, is strongly convex in the Euclidean sense. Intuitively, we can expect that such a situation arises when the logarithmic map $\operatorname{Exp}_x^{-1}$ does not affect the shape of $\mathcal{C}$ too much, which is the case when the sectional curvature of $\mathcal{M}$ is bounded and the set $\mathcal{C}$ is not too large.

In this section, we show that some sublevel sets of the function $x \mapsto \frac{1}{2} d_{\mathcal{M}}(x, x_0)^2$ are Riemannian strongly convex. Before proving the main theorem, we first introduce some concepts and two known results: **1)** distance functions are locally geodesically smooth and strongly convex (Martínez-Rubio and Pokutta, 2022), and **2)** locally, Euclidean pulled-back functions of geodesically smooth, strongly convex functions are smooth and strongly convex in the Euclidean sense (Criscitiello and Boumal, 2021).

## D.1 BOUNDED CURVATURE

In the following, we make an assumption regarding the curvature of our manifolds. Let us recall that, given a *two*-dimensional subspace $V \subseteq T_x \mathcal{M}$ of the tangent space of a point $x$, the sectional curvature at $x$ with respect to $V$ is defined as the Gauss curvature for the surface $\operatorname{Exp}_x(V)$ at $x$. The Gauss curvature at a point $x$ can be defined as the product of the maximum and minimum curvatures of the curves resulting from intersecting the surface with planes that are normal to the surface at $x$. See more details on the curvature tensor $\mathfrak{R}$ in (Petersen, 2006). Our assumption is as follows.

**Assumption D.1.** *The sectional curvatures of $\mathcal{M}$ are contained in the interval $[\kappa_{\min}, \kappa_{\max}]$ and the covariant derivative of the curvature tensor is bounded as $\|\nabla \mathfrak{R}\| \le F$.*

This assumption is not overly restrictive. The majority of the applications of Riemannian optimization are in locally symmetric spaces, which satisfy $\nabla \mathfrak{R} = 0$, for instance, constant curvature spaces, the SPD matrix manifold with the usual metric, $SO(n)$, and the Grasmannian manifold (Lezcano-Casado, 2020).

## D.2 STRONG CONVEXITY AND SMOOTHNESS OF DISTANCE FUNCTIONS

We now state a fact regarding the smoothness and strong convexity of the distance squared to a point, that is central to many Riemannian optimization algorithms. In the sequel, we use the notation $K \stackrel{\text{def}}{=} \max\{|\kappa_{\min}|; \kappa_{\max}\})$.

**Proposition D.2.** *(See (Martínez-Rubio and Pokutta, 2022)) Let us consider a uniquely geodesic Riemannian manifold $\mathcal{M}$ of sectional curvature bounded in $[\kappa_{\min}, \kappa_{\max}]$ and a ball $B_{x_0}(r)$ in $\mathcal{M}$ of radius $r$ centered at $x_0$. The function $x \mapsto \frac{1}{2} d_{\mathcal{M}}(x, x_0)^2$ is then $\delta_r$-strongly convex and $\zeta_r$-smooth in $B_{x_0}(r)$, where $\delta_r$ and $\zeta_r$ are the geometric constants defined by*

$$\zeta_r \stackrel{\text{def}}{=} \begin{cases} r\sqrt{|\kappa_{\min}|} \coth(r\sqrt{|\kappa_{\min}|}) & \text{if } \kappa_{\min} \le 0 \\ 1 & \text{if } \kappa_{\min} > 0 \end{cases}; \quad \delta_r \stackrel{\text{def}}{=} \begin{cases} 1 & \text{if } \kappa_{\max} \le 0 \\ r\sqrt{\kappa_{\max}} \cot(r\sqrt{\kappa_{\max}}) & \text{if } \kappa_{\max} > 0 \end{cases} \tag{41}$$

In the case of Cartan-Hadamard manifolds, $\delta_r = 1$ and $\zeta_r \in [r\sqrt{|\kappa_{\min}|}, r\sqrt{|\kappa_{\min}|} + 1]$.

## D.3 SMOOTHNESS AND STRONG CONVEXITY OF EUCLIDEAN PULLED-BACK FUNCTION

Under Assumption D.1, (Criscitiello and Boumal, 2021, Proposition 6.1) showed that the Euclidean pulled-back function $x \mapsto f(\operatorname{Exp}_{x_0}(x))$ of a smooth, strongly convex function $f$ defined in a ball of restricted radius in $\mathcal{M}$ is also smooth, strongly convex (in the Euclidean sense) in a ball of the same radius.

**Proposition D.3.** *(Informal, see (Criscitiello and Boumal, 2021, Proposition 6.1) for details). Let $\mathcal{M}$ be a uniquely geodesic Riemannian manifold that satisfies Assumption D.1 and that contains*

$\mathcal{B}(x_{ref}, r)$ *defined as*

$$\mathcal{B}(x_{ref}, r) : \{x \in \mathcal{M} : d_{\mathcal{M}}(x, x_{ref}) \leq r\}, \qquad \text{and let} \qquad \mathcal{B}_{x_{ref}}(0, r) : \{v \in \mathcal{T}_{x_{ref}}\mathcal{M} : \|v\| \leq r\}.$$

*As shown above, we also defined the pulled-back ball to $\mathcal{T}_{x_{ref}}\mathcal{M}$. Let us assume the function $f : \mathcal{M} \to \mathbb{R}$ is $L$-smooth, $\mu$-strongly geodesically convex in the ball $\mathcal{B}(x_{ref}, r)$, and has its minimizer $x^{\star} \in \mathcal{B}(x_{ref}, r)$. Then, if*

$$r \leq \frac{\mu}{L} \min\left\{\frac{1}{4K}, \frac{K}{4F}\right\},$$

*the pulled-back Euclidean function $x \mapsto f(\mathrm{Exp}_{x_0}(x))$ is $\frac{3}{2}L$-smooth, and $\frac{1}{2}\mu$-strongly convex over the ball $\mathcal{B}_{x_{ref}}(0, r)$.*

One can relax the assumption of the minimizer being in the ball as this is only used to show that the Lipschitz constant of the function in the ball is at most $2Lr$. For instance, if the distance between the minimizer and $x_{\mathrm{ref}}$ is $R$, the Lipschitz constant can be bounded by $O(LR)$, and we could use this bound to conclude a similar statement.

## D.4 MAIN RESULT

Using Propositions D.2 and D.3, we can prove that balls and other sets obtained from sublevel sets of smooth and strongly g-convex functions in small regions are Riemannian strongly convex.

**Theorem D.4.** *Let $\mathcal{M}$ be a uniquely geodesic Riemannian manifold that satisfies Assumption D.1, and let the function $f : x \mapsto \frac{1}{2}d_{\mathcal{M}}(x, x_0)^2$, $x_0 \in \mathcal{M}$. Then, the sublevel sets*

$$\mathcal{C}_r = \{x \in \mathcal{M} : f(x) \leq \frac{1}{2}r^2\}$$

*are Riemannian strongly convex if $r \leq \frac{1}{2}\frac{\delta_r}{\zeta_r}\min\{\frac{1}{4K}, \frac{K}{4F}\}$, where $\zeta_r$ and $\delta_r$ are the smoothness and strong convexity parameters of the function $\frac{1}{2}d_{\mathcal{M}}(\cdot, x_0)^2$ in $\mathcal{C}_r$ (see Eq. (41)).*

This implies that Riemannian balls in these generic manifolds are geodesically strongly convex sets. We note that the greatest $r$ that satisfies the condition in Theorem D.4 is roughly a constant, considering constant curvature bounds. Being able to optimize in these sets is important as Martínez-Rubio (2020) proved that one can reduce global geodesically convex optimization to the optimization over these sets.

**Impact of the space curvature** Curvature enters our results only through (i) the convexity radius of the manifold and (ii) bounds ensuring the exponential/log map distorts norm distances by controlled factors. These constants influence only the strong convexity modulus $\alpha$ but do not affect the qualitative nature of our results. Any manifold used in practice (SPD, Stiefel, Grassmann, sphere, hyperbolic models) has bounded curvature on compact regions; hence the assumptions of Theorem D.4 and Assumption 1.3 always hold for feasible sets of sufficiently small diameter.

## E   EXAMPLE OF A SIMPLE LINEAR MINIMIZATION ORACLE

When the curvature is constant (spheres or hyperbolic spaces) and the domain is a ball, the linear minimization oracle can be simplified into a one-dimensional problem.

**Theorem E.1** (**Linear minimization oracle in a ball in a constant curvature manifold**). *Let $\bar{B} \stackrel{\text{def}}{=} \mathcal{B}(x_0, r) \subset \mathcal{M}$ be a closed Riemannian ball in a manifold $\mathcal{M}$ of a constant sectional curvature. If the curvature $K$ is positive, let us assume $r < \frac{\pi}{2\sqrt{K}}$, so that the ball is uniquely geodesically convex. Given a point $x$ and a direction $v \in T_x\mathcal{M}$, we define*

$$\Gamma \stackrel{\text{def}}{=} \text{Exp}_x \left( \text{span}\{\text{Exp}_x^{-1}(x_0), v\} \right) \cap \{x \, | d_{\mathcal{M}}(x, x_0) = r\}.$$

*The solution of $\text{argmin}_{z \in \bar{B}} \langle v, \text{Exp}_x^{-1}(z) \rangle$ can then be found in $\Gamma$.*

A direct consequence of Theorem E.1 is that the problem of approximating a solution of the linear max oracle can be solved by solving an alternative one-dimensional problem.

**Proposition E.2.** *Let $u_1$, $u_2$ be an orthonormal basis for the linear subspace $\text{span}\{\text{Exp}_x^{-1}(x_0), v\}$.*

*The solution of $\text{argmin}_{z \in \bar{B}} \langle v, \text{Exp}_x^{-1}(z) \rangle$ can then be obtained by solving the following one-dimensional problem,*

$$\min_{\phi \in [-\pi, \pi]} \alpha(\phi) \langle v; p(\phi) \rangle,$$

*where $p(\phi)$ and $\alpha(\phi)$ over $\phi \in [-\pi, \pi]$ as*

$$p(\phi) = \cos(\phi)u_1 + \sin(\phi)u_2, \quad \alpha(\phi) = \underset{\alpha > 0}{\text{argmin}} : \text{Dist}\left(\text{Exp}_x(\alpha p(\phi)), x_0\right) = r. \tag{42}$$

The solution $\phi$ can be computed using a one-dimensional solver, e.g., bisection or the Newton-Raphson method. It should be noted that finding $\alpha(\phi)$ is also a one-dimensional problem, and it can also be solved using similar techniques. In some cases, there are simple formulas for $\alpha(\phi)$, e.g., when the manifold is a sphere.

**Proposition E.3.** *Let $\mathcal{M} = \mathbb{S}^{n-1}$ be the unit sphere. Then, $\alpha(\phi)$ in Eq. (42) reads*

$$\alpha(\phi) = 2\tan^{-1}\left( \frac{b(\phi) + \sqrt{a^2 + b(\phi)^2 - c^2}}{a + c} \right), \quad a = x_0^T x, \quad b(\phi) = x_0^T p(\phi), \quad c = 1 - 2\sin\left(\frac{r}{2}\right).$$

# F  RFW AND LINEAR MINIMIZATION ORACLE ON THE SDP MANIFOLD

We now present the derivation of the Riemannian Frank-Wolfe (RFW) update for the case where the feasible region is a geodesic ball in the SPD manifold equipped with the affine-invariant metric. Throughout this section we drop iteration subscripts when unambiguous.

Given the current iterate $X \in \mathcal{S}_{++}^n$ and gradient $G \in T_X \mathcal{S}_{++}^n$, the linear minimization oracle (LMO) associated with the trust region

$$\mathcal{B}(X, r) := \{V \in \mathcal{S}_{++}^n : d(X, V) \leq r\}$$

is the problem

$$\min_{V \in \mathcal{S}_{++}^n} \ \langle G, \log_X(V) \rangle_X \qquad \text{s.t.} \qquad d(X, V) \leq r. \tag{43}$$

At first sight, problem (43) appears difficult: the feasible set is curved and the objective depends on the Riemannian logarithm. However, we show below that the LMO admits a simple one–dimensional parametrization. In particular, every optimal solution lies in a two–dimensional subspace spanned by $\{\log_X(X_0), G\}$.

## F.1  THE SDP MANIFOLD WITH AFFINE-INVARIANT GEOMETRY

We first define the SPD manifold with affine-invariant geometry below.

**Definition F.1** (SPD manifold with affine-invariant geometry). *The space of symmetric positive definite matrices*

$$\mathcal{S}_{++}^n = \{X \in \mathbb{R}^{n \times n} : X = X^\top, \ X \succ 0\}$$

*forms a smooth Riemannian manifold when endowed with the affine-invariant metric. For any $X, Y \in \mathcal{S}_{++}^n$ and any $\Delta, U, V \in T_X \mathcal{S}_{++}^n$, the associated operators are:*

$$d(X, Y) = \left\| \log(X^{-1/2} Y X^{-1/2}) \right\|_F, \tag{44}$$

$$\log_X(Y) = X^{1/2} \log\left(X^{-1/2} Y X^{-1/2}\right) X^{1/2}, \tag{45}$$

$$\exp_X(\Delta) = X^{1/2} \exp\left(X^{-1/2} \Delta X^{-1/2}\right) X^{1/2}, \tag{46}$$

$$\langle U, V \rangle_X = \left\langle X^{-1/2} U X^{-1/2}, \ X^{-1/2} V X^{-1/2} \right\rangle = \operatorname{Tr}\left(X^{-1} U X^{-1} V\right). \tag{47}$$

The main advantage of this geometry is its invariance under congruent transformations from the set

$$GL(n) \ = \ \{A \in \mathbb{R}^{n \times n} \ : \ \det(A) \neq 0\},$$

**Theorem F.2** (Affine invariance of the SPD metric). *Let $\mathcal{S}_{++}^n = \{X \in \mathbb{S}^n : X \succ 0\}$ and*

$$\langle U, V \rangle_X \ = \ \operatorname{tr}\left(X^{-1} U X^{-1} V\right), \qquad X \in \mathcal{S}_{++}^n, \ U, V \in T_X \mathcal{S}_{++}^n.$$

*For $A \in GL(n)$, define the congruence action*

$$\Phi_A : \mathcal{S}_{++}^n \to \mathcal{S}_{++}^n, \qquad \Phi_A(X) = A X A^\top.$$

*Then $\Phi_A$ is an isometry for this metric, i.e.*

$$\langle d\Phi_A(X)[U], \ d\Phi_A(X)[V] \rangle_{\Phi_A(X)} = \langle U, V \rangle_X$$

*for all $X \in \mathcal{S}_{++}^n$ and all $U, V \in T_X \mathcal{S}_{++}^n$.*

*Proof.* Since $\Phi_A$ is linear, its differential satisfies

$$d\Phi_A(X)[U] = A U A^\top.$$

Also, $(A X A^\top)^{-1} = A^{-\top} X^{-1} A^{-1}$. Hence

$$\langle A U A^\top, \ A V A^\top \rangle_{A X A^\top} = \operatorname{tr}\left((A X A^\top)^{-1}(A U A^\top)(A X A^\top)^{-1}(A V A^\top)\right)$$

$$= \operatorname{tr}\left(A^{-\top} X^{-1} U X^{-1} V A^\top\right).$$

By cyclicity of the trace,

$$\mathrm{tr}\big(A^{-\top}MA^{\top}\big) = \mathrm{tr}(M), \qquad M = X^{-1}UX^{-1}V,$$

so

$$\langle \mathrm{d}\Phi_A(X)[U],\, \mathrm{d}\Phi_A(X)[V]\rangle_{\Phi_A(X)} = \mathrm{tr}\left(X^{-1}UX^{-1}V\right) = \langle U, V\rangle_X.$$

$\square$

As a consequence, the distance function is also invariant under congruent transforms.

**Corollary F.3** (Affine invariance of the distance). *Let $d(X, Y)$ be the geodesic distance induced by $\langle\cdot,\cdot\rangle_X = \mathrm{tr}(X^{-1}UX^{-1}V)$. For any $A \in GL(n)$,*

$$d\left(AXA^{\top},\, AYA^{\top}\right) = d(X, Y).$$

*Proof.* For any smooth curve $\gamma(t)$ joining $X$ to $Y$, define $\tilde{\gamma}(t) = A\gamma(t)A^{\top}$. By metric invariance,

$$\|\dot{\tilde{\gamma}}(t)\|_{\tilde{\gamma}(t)} = \|\dot{\gamma}(t)\|_{\gamma(t)}.$$

Thus $\mathrm{Length}(\tilde{\gamma}) = \mathrm{Length}(\gamma)$. Taking the infimum over all curves gives

$$d(AXA^{\top},\, AYA^{\top}) = d(X, Y).$$

$\square$

**Theorem F.4** (Equivariance of the exponential map under congruence). *Let $\mathcal{S}^n_{++}$ be endowed with the affine-invariant metric*

$$\langle U, V\rangle_X = \mathrm{tr}\big(X^{-1}UX^{-1}V\big), \qquad X \in \mathcal{S}^n_{++},\, U, V \in T_X\mathcal{S}^n_{++}.$$

*For any $A \in GL(n)$, consider the congruence action*

$$\Phi_A : \mathcal{S}^n_{++} \to \mathcal{S}^n_{++}, \qquad \Phi_A(X) = AXA^{\top}.$$

*Then the Riemannian exponential satisfies the equivariance identity*

$$\exp_{\Phi_A(X)}\big(\mathrm{d}\Phi_A(X)[U]\big) = \Phi_A\big(\exp_X(U)\big), \qquad X \in \mathcal{S}^n_{++},\, U \in T_X\mathcal{S}^n_{++}.$$

*Equivalently,*

$$\exp_{AXA^{\top}}\big(AUA^{\top}\big) = A\,\exp_X(U)\,A^{\top}.$$

*Proof.* Since $\Phi_A$ is an isometry (Theorem F.2), it maps geodesics to geodesics while preserving their speeds. Let $\gamma(t)$ be the unique geodesic with initial data

$$\gamma(0) = X, \qquad \dot{\gamma}(0) = U, \qquad \gamma(1) = \exp_X(U).$$

Define $\tilde{\gamma}(t) = \Phi_A(\gamma(t)) = A\gamma(t)A^{\top}$. By isometry,

$$\tilde{\gamma}(0) = AXA^{\top}, \qquad \dot{\tilde{\gamma}}(0) = AUA^{\top}, \qquad \tilde{\gamma}(t) \text{ is a geodesic.}$$

Uniqueness of geodesics with prescribed initial conditions implies

$$\tilde{\gamma}(t) = \exp_{AXA^{\top}}\big(t\,AUA^{\top}\big).$$

Evaluating at $t = 1$,

$$A\exp_X(U)A^{\top} = \tilde{\gamma}(1) = \exp_{AXA^{\top}}\big(AUA^{\top}\big).$$

$\square$

### F.2 A One-Dimensional Reduction for the LMO in an SPD Ball

We now show that the LMO can be reduced to a one dimensional problem. First, the proposition below parametrize the solution of the LMO $V^*$ by $V^* = \mathrm{Exp}_{X_k} \alpha(p^*)p^*$, where $p^*$ is a unit norm vector.

**Proposition F.5** (Directional reduction of the LMO). *For any unit tangent direction $p \in T_{X_k}\mathcal{S}_{++}^n$ with $\|p\|_{X_k} = 1$, define $\alpha(p) > 0$ as the unique solution of*

$$d\left(X_0, \ \exp_{X_k}\left(\alpha(p)\,p\right)\right) = r.$$

*Then every solution of the LMO lies in the set*

$$\Gamma = \left\{ \exp_{X_k}\left(\alpha(p)\,p\right) \ : \ p \in T_{X_k}\mathcal{S}_{++}^n, \ \|p\|_{X_k} = 1 \right\}.$$

*Moreover,*

$$V^\star = \underset{\|p\|_{X_k}=1}{\mathrm{argmin}} \ \alpha(p) \left\langle G_k, \, p\right\rangle_{X_k}.$$

*Proof.* Since the objective $V \mapsto \langle G_k, \log_{X_k}(V)\rangle_{X_k}$ is smooth and $C$ is compact, an optimal solution exists. If $V^\star$ were in the interior of $C$, a sufficiently short geodesic move along $(-\mathrm{grad})$ would decrease the objective while remaining feasible, contradicting optimality. Thus $d(X_0, V^\star) = r$.

For any unit direction $p \in T_{X_k}\mathcal{S}_{++}^n$, the function $\alpha \mapsto d\big(X_0, \exp_{X_k}(\alpha p)\big)$ is strictly increasing, hence it intersects the boundary $d(\cdot, X_0) = r$ at a unique $\alpha(p) > 0$. Therefore every boundary point may be written as $\exp_{X_k}(\alpha(p)p)$ for some $p$, establishing the representation of $\Gamma$.

Along each such geodesic ray,

$$\left\langle G_k, \log_{X_k}(\exp_{X_k}(\alpha p))\right\rangle_{X_k} = \alpha \left\langle G_k, p\right\rangle_{X_k},$$

so minimizing over the boundary reduces to minimizing $\alpha(p)\langle G_k, p\rangle_{X_k}$ over unit vectors $p$. $\qquad\square$

**Proposition F.6** (Reduction to a two-dimensional subspace.). *In Proposition F.5, the direction $p$ belong to the subspace*

$$p \in \mathrm{span}\{\log_{X_k}(X_0), \, G_k\}.$$

*Proof.* Let $W := \mathrm{span}\{\log_{X_k}(X_0), G_k\} \subset T_{X_k}\mathcal{S}_{++}^n$. For any unit tangent vector $p \in T_{X_k}\mathcal{S}_{++}^n$ we decompose

$$p = p_W + p_\perp, \qquad p_W \in W, \quad p_\perp \in W^\perp, \qquad \|p\|_{X_k} = 1.$$

We need to show that (i) $\alpha(p) = \alpha(p_W)$ and (ii) $\langle G_k, p\rangle_{X_k} = \langle G_k, p_W\rangle_{X_k}$.

*(i) The boundary constraint depends only on the angle with $\log_{X_k}(X_0)$.* Because the affine-invariant metric is preserved under congruence transformations, so is its induced distance:

$$d(AXA^\top, AYA^\top) = d(X, Y) \qquad \forall A \in GL(n).$$

Let $X_k \in \mathcal{S}_{++}^n$ and define

$$Z := X_k^{-1/2} X_0 X_k^{-1/2}.$$

Take an eigen-decomposition

$$Z = Q \exp(\Lambda) Q^\top,$$

with $Q$ orthogonal and $\Lambda$ diagonal. Then set

$$A := Q^\top X_k^{-1/2}.$$

This $A$ satisfies

$$AX_kA^\top = I, \qquad AX_0A^\top = \exp(\Lambda).$$

In these coordinates the geodesic from $I$ in direction $p$ is $t \mapsto \exp(t\tilde{p})$ with $\tilde{p} = ApA^\top$, and the distance to $X_0$ becomes

$$d\left(X_0, \exp_{X_k}(tp)\right) = d\left(\exp(\Lambda), \exp(t\tilde{p})\right).$$

For the affine-invariant metric, distances from $\exp(\Lambda)$ depend only on the *radial component* of $\tilde{p}$ in the direction of $\Lambda$, i.e. on the angle between $\tilde{p}$ and $\Lambda$. All components of $\tilde{p}$ orthogonal to $\Lambda$ contribute only tangentially and do not change $d(\exp(\Lambda), \exp(t\tilde{p}))$. Thus the map

$$t \;\mapsto\; d\left(X_0, \exp_{X_k}(tp)\right)$$

is completely determined by the angle between $p$ and $\log_{X_k}(X_0)$, and is strictly increasing in $t$.

Since $p$ and its projection $p_W$ have the same angle with $\log_{X_k}(X_0)$, they induce the same boundary-hitting parameter:

$$\alpha(p) = \alpha(p_W).$$

*(ii) The objective only depends on the projection onto* $\mathrm{span}\{G_k\}$. Since $p_\perp$ is orthogonal to $G_k$ in the Riemannian metric,

$$\langle G_k, p \rangle_{X_k} = \langle G_k, p_W \rangle_{X_k}.$$

Combining (i) and (ii), for every feasible $p$,

$$\alpha(p)\,\langle G_k, p \rangle_{X_k} = \alpha(p_W)\,\langle G_k, p_W \rangle_{X_k}.$$

Hence any direction $p$ is no better than its projection $p_W$, and no minimizer can have a component outside $W$. The search can therefore be restricted without loss of optimality to the two-dimensional subspace $W$. $\qquad\square$

This yields a practical one-dimensional formulation.

**Theorem F.7** (Angular parametrization of the LMO). *Let*

$$W := \mathrm{span}\{\log_{X_k}(X_0),\, G_k\} \subset T_{X_k}\mathcal{S}^n_{++}$$

*and choose an orthonormal basis $\{u_1, u_2\}$ of $W$. For $\phi \in [-\pi, \pi]$ define*

$$p(\phi) := \cos(\phi)\,u_1 + \sin(\phi)\,u_2, \qquad \|p(\phi)\|_{X_k} = 1,$$

*and let $\alpha(\phi) > 0$ be the unique scalar such that*

$$d\left(X_0, \exp_{X_k}\left(\alpha(\phi)\,p(\phi)\right)\right) = r.$$

*Then, solving the LMO Eq. (43)*

$$\min_{V \in \mathcal{S}^n_{++}} \langle G, \log_X(V) \rangle_X \qquad s.t. \qquad d(X, V) \leq r. \tag{48}$$

*is equivalent to solve the one-dimensional problem*

$$\min_{\phi \in [-\pi, \pi]} \alpha(\phi)\,\big\langle G, p(\phi) \big\rangle_{X_k},$$

*and then form $V^* = \alpha(\phi^*)p(\phi^*)$.*

*Proof.* By Proposition F.6, any optimal direction $p$ lies in $W$, so we can write $p = p(\phi)$ for a unique $\phi \in [-\pi, \pi]$. By part (i) of that proposition, the boundary parameter $\alpha$ depends only on the angle with $\log_{X_k}(X_0)$, hence on $\phi$, giving $\alpha = \alpha(\phi)$. By part (ii), the objective depends only on the projection of $p$ onto $\mathrm{span}\{G_k\}$, hence

$$\alpha\,\langle G_k, p \rangle_{X_k} = \alpha(\phi)\,\big\langle G_k, p(\phi) \big\rangle_{X_k}.$$

Thus minimizing over all feasible $p$ is equivalent to minimizing over $\phi \in [-\pi, \pi]$, as claimed. $\qquad\square$

Both $\alpha(\phi)$ and the outer minimization over $\phi$ can be solved efficiently using one-dimensional solvers such as bisection or Newton's method.

## F.3 ALGORITHM FOR THE LMO

We present two equivalent versions of the LMO. The first (Algorithm 2) operates directly in the affine-invariant geometry. The second (Algorithm 3) uses a congruence transform to simplify all expressions.

Using affine invariance (Theorem F.2), introduce the normalized variables

$$\tilde{X} = X_k^{-1/2} X X_k^{-1/2}, \qquad \tilde{X}_0 = X_k^{-1/2} X_0 X_k^{-1/2}.$$

In these coordinates we have

$$\tilde{X}_k = I, \qquad \tilde{X}_0 = \exp(\Lambda),$$

for some symmetric matrix $\Lambda$. At $I$, the affine-invariant metric coincides with the Frobenius structure:

$$\langle U, V \rangle_I = \mathrm{Tr}(U^\top V), \qquad \|U\|_I = \|U\|_F.$$

Moreover,

$$\log_I(Y) = \log(Y), \qquad \exp_I(S) = \exp(S).$$

In this normalized geometry, all Riemannian quantities at $X_k$ reduce to standard matrix exponentials, logarithms, and Frobenius inner products. The LMO therefore becomes a two–parameter search $(\phi, \alpha)$ over symmetric matrices of unit Frobenius norm, yielding the simplified algorithm in Algorithm 3.

---

**Algorithm 2** LMO on an SPD ball (affine-invariant geometry)

---

**Require:** $X_0, X_k \in \mathcal{S}_{++}^n$, radius $r > 0$, gradient $G_k$
**Ensure:** $V^\star$

**(A) Construct the 2D subspace** $W$
1: $u_1 = \log_{X_k}(X_0); \quad u_1 \leftarrow u_1/\|u_1\|_{X_k}$
2: $g_\perp = G_k - \langle G_k, u_1 \rangle_{X_k} u_1; \quad u_2 = g_\perp/\|g_\perp\|_{X_k}$

**(B) Angular parametrization**
3: For $\phi \in [-\pi, \pi]$, set
$$p(\phi) = \cos\phi\, u_1 + \sin\phi\, u_2.$$

**(C) Reduced objective**
$$F(\phi) = \alpha(\phi)\, \langle G_k, p(\phi) \rangle_{X_k}, \qquad \alpha(\phi) \text{ solves } d\big(X_0, \exp_{X_k}(\alpha\, p(\phi))\big) = r.$$
More explicitly,
$$\alpha(\phi) \text{ solves } \left\| \log\!\Big( X_0^{-1/2} X_k^{1/2} \exp\!\big( \alpha\, X_k^{-1/2} p(\phi) X_k^{-1/2} \big) X_k^{1/2} X_0^{-1/2} \Big) \right\|_F = r.$$

**(D) Final 1D minimization**
$$\phi^\star = \arg\min_{\phi \in [-\pi, \pi]} F(\phi), \qquad p^\star = p(\phi^\star), \quad \alpha^\star = \alpha(\phi^\star).$$

4: **return** $V^\star = \exp_{X_k}(\alpha^\star p^\star)$

---

---

**Algorithm 3** LMO on an SPD ball in normalized coordinates

**Require:** $X_0, X_k \in \mathcal{S}_{++}^n$, radius $r > 0$, gradient $G_k$
**Ensure:** $V^\star$

1: **(Normalization)**
2: $M \leftarrow X_k^{-1/2} X_0 X_k^{-1/2}$
3: Compute eigendecomposition
$$M = Q \exp(\Lambda) Q^\top,$$
with $\Lambda$ diagonal (and symmetric).
4: Define the congruence:
$$A \leftarrow Q^\top X_k^{-1/2}.$$
Then
$$\tilde{X} = AXA^\top, \quad \tilde{X}_k = I, \quad \tilde{X}_0 = \exp(\Lambda).$$
5: Transform the gradient:
$$\tilde{G}_k = AG_k A^\top = Q^\top X_k^{-1/2} G_k X_k^{-1/2} Q.$$

    **(A) Construct the 2D subspace** $W$
6: $u_1 = \log(\tilde{X}_0) = \Lambda; \quad u_1 \leftarrow u_1/\|u_1\|_F.$
7: $g_\perp = \tilde{G}_k - \langle \tilde{G}_k, u_1 \rangle_F \, u_1; \quad u_2 = g_\perp/\|g_\perp\|_F.$

    **(B) Angular parametrization**
8: For $\phi \in [-\pi, \pi]$, define
$$p(\phi) = \cos\phi \, u_1 + \sin\phi \, u_2.$$

    **(C) Solve for** $\alpha(\phi)$
9: For each $\phi$, define $\alpha(\phi) > 0$ as the solution of
$$\left\| \log\big( \exp(-\Lambda/2) \exp(\alpha(\phi)\, p(\phi)) \exp(-\Lambda/2)\big)\right\|_F = r.$$
(This is a 1D root-finding problem.)
10: Define the reduced objective
$$F(\phi) = \alpha(\phi) \, \langle \tilde{G}_k, p(\phi) \rangle_F.$$

    **(D) Final 1D minimization**
$$\phi^\star = \arg\min_{\phi \in [-\pi, \pi]} F(\phi), \qquad p^\star = p(\phi^\star), \quad \alpha^\star = \alpha(\phi^\star).$$
11: Compute the normalized solution
$$\tilde{V}^\star = \exp(\alpha^\star p^\star).$$

12: **(E) Map back to original coordinates**
      Since $\tilde{X} = AXA^\top$ with $A = Q^\top X_k^{-1/2}$,
$$V^\star = A^{-1} \tilde{V}^\star A^{-\top} = X_k^{1/2} Q \, \tilde{V}^\star Q^\top X_k^{1/2}.$$

13: **return** $V^\star$.

---

## F.4 Numerical Experiment: Minimization Over an SPD Ball

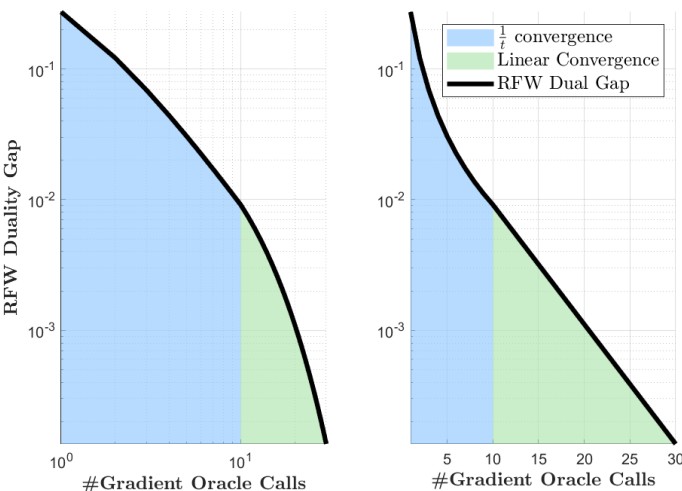

Figure 3: Convergence of the RFW algorithm (Algo. 1) for minimizing a geodesically smooth convex function over a geodesic ball in the SPD manifold endowed with the affine-invariant metric. The dimension is $n = 50$, the center $X_0$ is random, the radius is $r = 0.1$, and the objective is $f(X) = \frac{1}{2}d(X, Y)^2$ with $Y$ placed just outside the ball at distance $1.01 \times r$ (the closer to $r$, the harder the instance). As predicted by Theorem 5.2, the iterates display a clear transition from the global $O(t^{-1})$ regime to a locally linear rate. Interestingly, the curve suggests the possibility of a global linear rate for this particular geometry (SPD with the affine-invariant metric), as the observed behavior matches the minimum of the two theoretical regimes.

This experiment tests the convergence guarantees of Theorem 5.2 on the SPD manifold $\mathcal{S}{++}^n$ endowed with the affine-invariant geometry. We consider the problem

$$\min_{X \in \mathcal{S}_{++}^n} f(X) \qquad \text{s.t.} \qquad d(X, X_0) \leq r, \tag{49}$$

where $f(X) = \frac{1}{2}d(X, Y)^2$ and $Y$ is chosen such that $d(X_0, Y) > r$, forcing the optimizer to lie on the boundary of the feasible set. The resulting convergence curve in Fig. 3 displays the two characteristic regimes: a global sublinear phase followed by a sharp transition to the linear rate ensured by the strong convexity of the SPD ball. This benchmark validates the theoretical predictions and confirms that the geometric structure of the feasible region governs the accelerated behavior of RFW.

## G MATLAB CODE FOR THE EXPERIMENTS

To run the experiment, it is required to have a sufficiently recent version of Matlab, and the toolbox Manopt (Boumal et al., 2014).

### G.1 SUBROUTINE: LINEAR MAX ORACLE FOR A BALL IN A SPHERE

```matlab
function v = linear_max_oracle(w, x, R, x0, manifold)

% Solve the problem
% max w^Tz : z \in exp_x^{-1}(C),
% where C := {x:dist(x,x_0)\leq R}
% Assume that the sphere has radius = 1!

% Define the two basis vector of the subspace u_1 and u_2
u1 = w;
u1 = u1/norm(u1);
u2 = manifold.log(x, x0);
u2 = u2-u1*(u1'*u2);

if norm(u2) > 0
    u2 = u2/norm(u2);
    p = @(phi) cos(phi)*u1 + sin(phi)*u2;

    alpha = @(p) solve_sincoseq(x0'*x,x0'*p,(1-2*sin(R/2)^2));

    fhandle = @(phi) -(alpha(p(phi))*cos(phi)); %same minimum
        since w*u2 = 0.
    options = optimset('TolX',eps);
    [best_phi] = fminbnd(fhandle, -pi, pi, options);
    v = manifold.exp(x, p(best_phi), alpha(p(best_phi)));
else
    alpha = solve_sincoseq(x0'*x,x0'*u1,(1-2*sin(R/2)^2)); % best
        direction: u1
    v = manifold.exp(x, u1, alpha);
end
```

### G.2 SAMPLE SCRIPT: RANDOM QUADRATIC OVER A SPHERE WITH BALL CONSTRAINT

```matlab
% Generate the problem data.
d = 50;
n = 25; % Slower convergence when n < d -> non-strongly convex
    case
nIter = 500;
dualgap = zeros(1, nIter);

% Manifold: this code uses manop, see https://www.manopt.org
manifold = spherefactory(d);

% Define the problem cost function and its derivatives.
A = randn(n,d);
A = A'*A;
A = A/norm(A);
xstar = manifold.rand();
f = @(x) 0.5*(x-xstar)'*A*(x-xstar);
mgrad = @(x) manifold.egrad2rgrad(x, A*(x-xstar));
L = norm(A); % Worst case

% Create the problem set
x_center = manifold.rand();
radius_ratio = 0.9; % <1: xstar is oustide
radius_max = manifold.dist(x_center, xstar)*radius_ratio;
setFunction = @(x) manifold.dist(x_center, x)<=radius_max;

x = x_center;
% Main loop RFW
for i=1:(nIter-1)
    gradx = mgrad(x);
    v = linear_max_oracle(-gradx, x, radius_max, x_center,
        manifold);
    dualgap(i) = -manifold.inner(x, gradx, manifold.log(x, v));

    step_size = -manifold.inner(x, gradx, manifold.log(x, v)) / (L
        *manifold.dist(x, v)^2);
    step_size = min(step_size, 1);
    x = manifold.exp(x, manifold.log(x, v), step_size);
end
v = linear_max_oracle(-gradx, x, radius_max, x_center, manifold);
dualgap(end) = -manifold.inner(x, gradx, manifold.log(x, v));

% Max at eps, otherwise the result is numerically meaningless
semilogy(1:length(dualgap), max(dualgap, eps))
legend({'FW Dual Gap'})
```

### LLM USAGE

The authors acknowledge the use of a large language model (LLM) to assist with improving the clarity and presentation of the manuscript.

