# OpenReview forum: "Strongly Convex Sets in Riemannian Manifolds"
_ICLR.cc/2026/Conference — ICLR 2026 Poster_

### Official Review · Reviewer_ArMd · 2025-10-30

**Soundness:** 3
**Presentation:** 2
**Contribution:** 3
**Rating:** 4
**Confidence:** 4

**Summary:**

This paper proposes a Riemannian Frank–Wolfe (RFW) algorithm with linear convergence guarantees under a novel set of strong convexity assumptions for constraint sets on manifolds. It extends the classical Euclidean FW framework and introduces several definitions of strong convex sets tailored to geodesic settings. The theoretical development is comprehensive and the main results are sound.

**Strengths:**

1. The paper generalizes linear convergence guarantees of the FW method from Euclidean to Riemannian settings under appropriately defined curvature-aware convexity.

2. The paper provides formal geometric characterizations and includes theoretical examples of sets satisfying the RSI condition.

**Weaknesses:**

1. Although the paper motivates the RFW method via machine learning applications (e.g., Example 5.3), there are no real ML experiments. In particular, the neural network loss mentioned in Example 5.3 is not geodesically strongly convex, and the main convergence theorems do not apply to such cases. This weakens the practical relevance of the proposed method.

2. The three definitions of strongly convex sets—geodesic, Riemannian, and double-geodesic—are introduced in sequence, but are somewhat hard to digest. A clearer summary (e.g., a comparison table showing hierarchy, assumptions, and known examples) would greatly help readability.

3. The proposed notion of strong convexity (and RSI) is in contrast with proximal smoothness (prox-regularity) in Euclidean space, where the corresponding inequalities (see Davis, Drusvyatskiy, and Shi, SIAM J. Optim. 2025) take the opposite form (i.e., bounding curvature from above instead of below). The paper should clarify that this is due to fundamentally different settings: the current work focuses on Riemannian geometry, while prox-regularity is established in embedded space.

references:
Davis, D., Drusvyatskiy, D., and Shi, Z. (2025). Stochastic Optimization Over Proximally Smooth Sets. SIAM Journal on Optimization, 35(1), 157–179.

**Questions:**

Which practical ML objectives/constraints on common manifolds (e.g., sphere, SPD, Grassmann) actually satisfy your assumptions (geodesic convexity + RSI, or approximate RSI + gradient lower bound)? Please provide at least one end-to-end experiment and a concrete example to check/estimate the RSI constant and strong-convexity modulus on real data.

---

> ### Author Response · Authors · 2025-11-17
>
> We thank the reviewer again for their careful reading and insightful comments. We hope that the responses below clarify the concerns that have been raised. If our explanations address the reviewer’s points, we would greatly appreciate it if the reviewer could let us know whether this affects their overall assessment.
>
> ---
>
> ## **Response to Weakness 1: “No real ML experiments / RFW motivation unclear”**
>
> There is a misunderstanding, since our paper does not aim to motivate or introduce the RFW algorithm. That contribution belongs to Weber & Sra (2023).
>
> Our contribution is the **geometric characterization of strongly convex sets in Riemannian manifolds**, which is independent of the objective function. Hence, example 5.3 is a relevant example of an ML problem that uses a strongly convex constraint over a manifold. Moreover, by making use of the reduction from Martinez-Rubio (2020) that we discussed, we can solve general smooth geodesically convex problems by solving them sequentially in balls of constant radius, and we showed that those balls are strongly convex sets for which our algorithm applies, making our algorithm available to a wide range of ML problems.
>
> The RFW example illustrates only why strong convexity is practically relevant to the optimization community.
>
> That said, in response to the reviewer’s request, we are currently developing an additional experiment on the SPD manifold, directly motivated by a discrete density-fitting problem arising in computational quantum physics. This experiment applies our theory in a setting substantially richer than the sphere and is more representative of real applications.
>
> We would appreciate clarification from the reviewer on whether such an experiment would address their concern.
>
>
> ---
>
> ## **Response to Weakness 2: “A clearer summary (e.g., a comparison table showing hierarchy, assumptions, and known examples) would greatly help readability.”**
>
> We are unsure what specific concern the reviewer has in mind, because the paper already provides:
>
> * **A complete implication graph** (Eq. 7) showing the hierarchy of the definitions;
> * **A geometric illustration** (Fig. 1) of the different notions;
> * **Numerous concrete examples**, including:
>
>   * Riemannian strongly convex sets (p. 5),
>   * Geodesic and double-geodesic strongly convex sets (p. 5),
>   * Sets on the sphere and SPD matrices (p. 7),
>   * Trust-region constraints (p. 8),
>   * Global Riemannian optimization constraints (p. 9),
>   * A machine-learning example with strongly convex constraints (Ex. 5.3),
>   * And a dedicated **Appendix D** giving further examples and sufficient conditions for Riemannian strongly convex sets
>
> However, if the reviewer believes there are specific parts of the paper that require clarification or additional examples, we invite the reviewer to mention these during the discussion period so that we can improve the paper's clarity.
>
> ---
>
> ## **Response to Weakness 3: “Relation to prox-regularity / opposite inequality direction”**
>
> We thank the reviewer for bringing this reference to our attention, which we were previously unaware of.
>
> While the two inequalities do involve related geometric quantities, they are in fact dual in spirit rather than equivalent. In particular:
>
> * The order is reversed: our Riemannian Scaling Inequality (RSI) provides a lower bound in the direction of the maximizer, whereas the uniform normal inequality provides an upper bound on all outward directions.
> * RSI is not uniform: we only require the inequality to hold at the argmax of the linear minimization oracle, which is crucial for Frank–Wolfe analysis. The uniform normal inequality requires the condition to hold for all directions y-x.
> * The geometric domains differ: RSI is formulated entirely in the tangent space (via the inverse exponential map), whereas the uniform normal inequality involves differences y−x in the ambient space, which may not lie on the manifold nor in the tangent space.
>
> Because of these differences, uniform normality and our RSI are best viewed as dual concepts: they are related but operate in opposite analytic directions and serve different algorithmic purposes. We will nevertheless mention this connection in the related work section to avoid any confusion and to clarify how the two notions differ.

---

> ### Author Response · Authors · 2025-12-01
> **Manuscript update**
>
> We thank again the reviewer for the review. Below we list the changes made to the manuscript in response to the reviewer’s comments. **The suggestion to add new numerical results was particularly valuable and led to a substantial strengthening of the manuscript.**
>
> - **Added a comparison table summarizing the three notions of strong convexity (Section 2).**
>   The table presents the  assumptions, and representative examples for geodesic, Riemannian, and double-geodesic strong convexity. This directly addresses the request for a clearer summary beyond the implication diagram.
>
> - **Added a new example on the SPD manifold (Section 5.4, appendix F).**
>   We included a practical density-fitting problem (finite-temperature free-energy minimization), which provides a concrete example of a strongly convex constraint set and illustrates the predicted linear-convergence behaviour. This lead to a new contribution (LMO on the SDP manifold) of approx. 7 pages that can be found in appendix F.
>
> - **Added discussion on relation to uniform normal inequality (Related Work).**
>   We now mention these inequalities and clarify that they are analytically dual to RSI: uniform normality is a global upper bound over all outward directions, while RSI is a localized lower bound at the LMO maximizer, tailored to Frank-Wolfe.
>
> These additions substantially strengthen the paper by improving clarity and providing a concrete practical example.

---

### Official Review · Reviewer_51us · 2025-11-03

**Soundness:** 3
**Presentation:** 2
**Contribution:** 3
**Rating:** 2
**Confidence:** 3

**Summary:**

The paper studies the notion of strong convexity in the setting of reimannian manifolds (and geodesic spaces). Though strong convexity is a well studied and natural notion in Euclidean space, generalization of this concept to manifolds is non trivial. The present paper studies various versions of such a definition and presents fast convergence of natural algorithms under these definitions. In particular, they show that a version of Frank-Wolfe converges linearly for strongly convex optimization

**Strengths:**

Generalization of the toolkit of optimization to the setting of manifolds has considered an interesting and important problem for the past several years, due to both theoretical and practical motivation. Systemizing Euclidean definition to the manifold setting is an important step and understanding what assumptions lead to fast algorithms is an important step

**Weaknesses:**

Though the problem is interesting, I personally think that the definitions presented in the paper are not particularly insightful and it is not clear how generally applicable and useful the defnition would be. As presented, the definition come off as a "syntactic" generalization of the Euclidean case and thus fail to convey insight into the RM optimization problem

**Questions:**

-> Further discussion of the assumptions are very important. For example, assumption 1.3 seems very strong to me. In particular, it is seems like a strong quantitative local Euclidean property that could potential make the difficultly of the RM setting a bit moot (I understand that there is still variation in the metric and this leads to technical difficulties; but it would be good discuss the "conceptual" hurdles.
-> Getting log(1/eps) rates for Geodesic optimization actually happens to be an important problem in subfields of theoretical computer science (eg https://arxiv.org/pdf/1910.12375) and has garnered interest in the field. Unfortunately most proposed algorithms work under assumptions that are hard to check or make assumptions the algorithm work only is "near Euclidean" settings. It would be interesting to see a discussion of the present paper in this context (or related applications)

---

> ### Author Response · Authors · 2025-11-17
>
> ## Response to Weakness: The definitions presented in the paper are not particularly insightful, and it is not clear how generally applicable and useful the definition would be [...] come off as a "syntactic" generalization of the Euclidean case and thus fail to convey insight into the RM optimization problem
>
> We respectfully disagree with the characterization that our definitions are merely syntactic generalizations. While they are inspired by Euclidean strong convexity, extending these notions to Riemannian manifolds is highly nontrivial.
>
> In Hilbert spaces, several notions of strong convexity (such as sublevel sets of smooth strongly convex functions, the existence of a ball around any segment inside the set, or the scaling inequality) are all equivalent. On manifolds, however, these notions are no longer comparable, leading to different definitions with distinct algorithmic implications. Refer to Eq. 7 for a diagram illustrating the relationships among these definitions.
>
> Moreover, we demonstrate that it is possible to define approximate versions of the scaling inequality that are weaker than the full Riemannian scaling inequality, yet still sufficient to guarantee linear convergence of algorithms such as the Frank-Wolfe algorithm. Importantly, this notion is genuinely nontrivial: it does not arise from a Euclidean setting, as it collapses to the exact scaling inequality in Euclidean spaces, highlighting that it captures intrinsic Riemannian geometric properties rather than being a mere translation of Euclidean concepts.
>
>
> ---
>
> ## Response to Question 1: Further discussion on assumptions. Assumption 1.3 seems very strong.
>
> The strongest of our assumptions is not strong, since we provide examples of Riemannian balls within a very general family of manifolds that satisfy the strongest of our notions. Importantly, it was shown that all geodesically convex problems can be reduced to a sequence of problems over these such balls, so our linearly convergent Frank-Wolfe algorithm can solve very general problems via this reduction.
>
> That being said, the paper does provide a discussion for most of its assumptions. However, as the reviewer pointed out, we do not discuss assumption 1.3.
>
> Given assumption 1.2, assumption 1.3 is not strong: Since the exponential map is bijective, the distance $d$ can be taken as exactly the norm of the exponential map, hence $\mu = L = 1$. We wanted a more general result where the distance tied to the manifold may not be the same as the distance that defines the smoothness and strong convexity of a function $f$; see Appendix C.4 for a detailed discussion.
>
> Additionally, assumption 1.2 is not overly restrictive either. This is a natural condition that extends the notion of convex sets from Hilbert spaces to manifolds. For instance, in the case of the sphere manifold, any geodesically convex subset that can be contained in a hemisphere is uniquely geodesic, but the whole sphere is not. This assumption is implicitly implied in the pioneer work on RFW from [Weber and Sra, 2023, Algorithm 2], as the inverse exponential is required to run the RFW algorithm.
>
> ---
>
>
> ## Response to Question 2: Most proposed algorithms work under assumptions that are hard to check, or make assumptions that the algorithm only works in "near Euclidean" settings.
>
> We provide practical tools to identify whether a set satisfies strong convexity in the Riemannian sense, making our assumptions checkable for commonly used manifolds. In fact, our framework already *provides similar verification tools to those in the Euclidean counterpart (Garber & Hazan, 2015)*. For instance,
> * Theorem 4.2 shows that if we can express (or construct) the set using sublevel sets of a geodesically smooth, strongly convex function, then the set is strongly convex.
> * Theorem D4 mentions that ball constraints with a certain radius are Riemannian strongly convex sets.
>
> Importantly, in our setting, *it is not necessary to run a modified version of Frank-Wolfe*. The algorithm automatically transitions to a linear convergence rate whenever the strong convexity conditions are satisfied, without any additional tuning. This behavior is analogous to gradient descent in Euclidean space: when the function is merely smooth and convex, the convergence rate is sublinear (1/k), but when it is strongly convex, the convergence becomes linear.
>
> ---
>
> Finally, while we sincerely appreciate the reviewer’s feedback, we are somewhat confused about how the qualitative points raised translate into the relatively low score of 2/10.
> We did emphasize our efforts to provide practical tools and clear algorithmic implications, so we were somewhat surprised by the comment regarding limited insight or applicability. We would appreciate it if the reviewer could provide concrete examples or more specific guidance on which definitions, assumptions, or results are perceived as lacking insight or applicability.

---

> > ### Author Response · Authors · 2025-12-01
> > **Manuscript update**
> >
> > We thank the reviewer for the review. Below we list the change made to the manuscript in response to the reviewer’s comments.
> >
> > - **Clarified Assumption 1.3 (Section 1).**
> >   We added an explicit explanation stating that under Assumption 1.2 (unique geodesics), Assumption 1.3 is mild: if chosen, the Riemannian distance coincides with the norm of the logarithmic map, so the constants can be taken as \( \ell_M = L_M = 1 \).
> >
> > This addition strengthen the discussion around assumptions and clarify the conceptual motivations behind the definitions, addressing the reviewer’s concerns directly within the manuscript. We did not modify the rest of the manuscript because the remaining concerns were conceptual rather than indicating gaps or inaccuracies in the exposition.

---

### Official Review · Reviewer_QN9G · 2025-11-03

**Soundness:** 3
**Presentation:** 3
**Contribution:** 2
**Rating:** 4
**Confidence:** 3

**Summary:**

This paper considers strongly convex sets in Riemannian manifolds and proposes several definitions, which generalize the Euclidean counterparts. In addition, the linear convergence rate of Riemannian FW algorithm can be derived via the proposed Riemannian scaling inequality condition.

**Strengths:**

Studying strongly convex sets in Riemannian manifolds is interesting, which requires a good understanding of both convex analysis and Riemannian optimization/geometry.

**Weaknesses:**

1. I did not see the practical improvement/implication of this paper's results on optimization over manifolds (e.g., SPD, SO(3), Stiefel manifolds, Grassmann manifolds). It would be great if the authors can explicitly provide some new results for concerte examples.
2. The numerical experiments are limited and only for sphere constraints. In this case, this paper is quite related to (https://arxiv.org/abs/1911.11955). In addition, there are several papers of optimization over SPD and Stiefel manifolds with linear convergence rate, e.g., (https://arxiv.org/abs/2001.01700, https://arxiv.org/abs/2112.06556). It would be better to compare with these papers.

**Questions:**

1. the scaling inequality is quite similar to the uniform normal inequality in https://www.heldermann-verlag.de/jca/jca02/jca02008.pdf and https://arxiv.org/pdf/2002.06309. It would be better to mention them in your paper.
2. Could you provide some concrete examples of $C$ that satisfy Assumption 1.2?
3. In lines 243-245, the authors mention "This theorem applies to most practical cases ...". Could the authors explain more about this part? I am curious about the implication of results in this paper on specific optimization over manifolds (e.g., SPD, SO(3), Stiefel manifolds).

---

> ### Author Response · Authors · 2025-11-17
>
> We hope the following explanation resolves the misunderstanding and clearly highlights why the cited works address fundamentally different problem settings. If any specific comparison was intended, we would be happy to clarify further, and we respectfully invite the reviewer to reconsider the paper’s assessment in light of the distinctions outlined below.
>
> ---
>
> ## Response to Weakness 1: “Practical improvement/implication on optimization over manifolds”
>
> There is a misunderstanding: the paper does provide concrete and practically meaningful implications for optimization over manifolds.
>
> First, we establish an explicit algorithmic improvement for the Riemannian Frank–Wolfe method. In Theorems 5.1 and 5.2, we demonstrate that under the strong convexity conditions of the set, the method achieves (local) linear convergence *even if the objective is not strongly convex*. This sharpens the best-known guarantees on widely used manifold domains such as the SPD cone, the sphere, the Stiefel manifold, and the Grassmann manifold, amongst others.
>
> Second, Section 5.2 presents two concrete applications that directly benefit from our results:
>
> * Example 1 (Trust-region subproblem): The trust-region method is one of the most widely used algorithms in Riemannian optimization. Our framework guarantees strong convexity of the feasible set, enabling faster convergence of standard trust-region solvers.
>
> * Example 2 (Global optimization via local optimization): Martinez-Rubio (2020) showed that global geodesically convex optimization can be reduced to a sequence of local problems posed on geodesic balls. Our results ensure that these local problems satisfy strong convexity, thereby yielding faster convergence rates for the entire global optimization scheme.
>
> Together, these results provide direct algorithmic improvements and practical consequences for manifold optimization.
>
>
>
> ---
>
> ## Response to Weakness 2a: “Numerical experiments are limited for sphere constraints”
>
> We intentionally avoided adding more involved numerical experiments, as doing so would risk obscuring the main theoretical message and further increasing the length and complexity of an already dense paper. Indeed, the primary contribution of the paper is the theoretical characterization of set-strong convexity on manifolds, and the Frank-Wolfe example was included merely as an illustration of how this characterization can lead to algorithmic improvements.
>
> That said, in response to the reviewer’s request, we are currently developing an additional experiment on the SPD manifold, directly motivated by a discrete density-fitting problem arising in computational quantum physics. This experiment applies our theory in a setting substantially richer than the sphere and is more representative of real applications.
>
> We would appreciate clarification from the reviewer on whether such an experiment would address their concern.
>
> ---
>
>
>
> ## Response to Weakness 2b: “In addition, there are several papers of optimization over SPD and Stiefel manifolds with linear convergence rate”
>
> The cited papers study optimization over the entire manifold (SPD, Stiefel, sphere) or smooth equality-constrained problems. In contrast, our paper analyzes optimization over a strongly convex subset of a manifold. This is the analogue of Euclidean optimization constrained over a convex body, and it is the setting where Frank-Wolfe-type methods are relevant.
>
> Because these works do not consider strongly convex feasible sets at all (their feasible domain is the entire manifold), they do not provide conditions, definitions, or guarantees related to the geometry of constraint sets. This is precisely what our paper contributes to the field. For this reason, we do not fully understand what type of comparison the reviewer expects, as the problem formulations, geometric assumptions, and algorithms are fundamentally different.
>
> Detailed differences with the cited works:
>
> (1911.11955) analyzes optimization on the sphere itself as the feasible domain (no strongly convex subset), using a structure specific to the entire manifold. It does not study strong convexity of subsets, nor does it address projection-free methods. In addition, it only focuses on minimizing quadratics.
>
> (2001.01700) studies Riemannian gradient descent on the entire SPD manifold and obtains linear rates under function-level assumptions (e.g., geodesic strong convexity). It does not consider constrained problems nor properties of feasible sets.
>
> (2112.06556) analyzes optimization on the Stiefel manifold with explicit retractions and smoothness assumptions. Again, the feasible region is the entire manifold; there is no notion of a strongly convex constraint set.

---

> ### Author Response · Authors · 2025-11-17
>
> ## Response to Question 1: “Scaling inequality is quite similar to uniform normal inequality”
>
> We thank the reviewer for bringing this reference to our attention, which we were previously unaware of.
>
> While the two inequalities do involve related geometric quantities, they are in fact dual in spirit rather than equivalent. In particular:
>
> * The order is reversed: our Riemannian Scaling Inequality (RSI) provides a lower bound in the direction of the maximizer, whereas the uniform normal inequality provides an upper bound on all outward directions.
> * RSI is not uniform: we only require the inequality to hold at the argmax of the linear minimization oracle, which is crucial for Frank-Wolfe analysis. The uniform normal inequality requires the condition to hold for all directions $y-x$.
> * The geometric domains differ: RSI is formulated entirely in the tangent space (via the inverse exponential map), whereas the uniform normal inequality involves differences $y-x$ in the ambient space, which may not lie on the manifold nor in the tangent space.
>
> Because of these differences, uniform normality and our RSI are best viewed as dual concepts: they are related but operate in opposite analytic directions and serve different algorithmic purposes. We will nevertheless mention this connection in the related work section to avoid any confusion and to clarify how the two notions differ.
>
>
>
> ---
>
> ## Response to Question 2: “Examples of C that satisfy Assumption 1.2”
>
> Assumption 1.2 states that the subset $C \subset \mathcal{M}$ is compact and uniquely geodesically convex.
>
> Here are several examples (the list is far from exhaustive):
> * **Convex subsets of Euclidean space.**  In $mathbb{R}^n$ with the standard metric, any compact convex set is uniquely geodesically convex since geodesics are straight-line segments.
> * **CAT(0) (Cartan-Hadamard) spaces.**  These spaces are globally uniquely geodesic. Hence, any compact **geodesically convex** subset of a CAT(0) space automatically satisfies the assumption.
> * **Subsets of the sphere contained in an open hemisphere.**  On $\mathbb{S}^n$, any geodesically convex subset lying entirely inside an open hemisphere is uniquely geodesically convex (ambiguities occur only at distances $\geq \pi/2$).
> * **Small geodesically convex sets on general Riemannian manifolds.**  On any Riemannian manifold with bounded curvature, there exists a sufficiently small neighborhood in which geodesics are unique and depend smoothly on endpoints. Any compact geodesically convex subset of such a neighborhood is therefore uniquely geodesically convex.
>
> Assumption 1.2 is closely related to the injectivity radius of the manifold. Within any region whose diameter is smaller than the local injectivity radius (or convexity radius), minimizing geodesics are unique.
>
> We also refer the reviewer to the examples in Riemannian Optimization via Frank-Wolfe Methods by Weber and Sra, which illustrate several optimization problems posed over subsets of manifolds that satisfy Assumption 1.2.
>
>
>
>
> ---
>
> ## Response to Question 3: “Could the authors explain more about this part "Theorem [D.4] applies to most practical cases ..."?”
>
> To clarify, we meant that Theorem D.4 applies specifically to manifolds with bounded curvature, which encompasses most manifolds commonly used in optimization, such as the SDP, SO(n), and Stiefel manifolds, among others.
>
> A key implication of our results arises when it is integrated with the framework of Martinez-Rubio (2020), who demonstrated that global geodesically convex optimization over these manifolds can be reduced to a sequence of local subproblems posed on geodesic balls. Within each such ball, our results establish strong convexity, thereby ensuring rapid convergence of standard optimization methods. Consequently, this local reduction, together with our analysis, yields improved convergence rates for all problem classes studied in Martinez-Rubio (2020).

---

> ### Author Response · Authors · 2025-12-01
> **Manuscript update**
>
> We thank the reviewer for the helpful feedback. **The suggestion to add new numerical results was particularly valuable and led to a substantial strengthening of the manuscript.**
>
> Below we list only the made to the manuscript in response to the reviewer’s comments.
>
> - **Added an SPD experiment (Section 5.4 and Appendix F).**
>   We introduced a new optimization example on the SPD manifold based on a finite-temperature free-energy model. This provides a concrete non-spherical case where our strong-convexity framework yields a clear linear-convergence regime. This lead to a new contribution (LMO on the SDP manifold) of approx. 7 pages that can be found in appendix F.
>
> - **Clarified applicability to common manifolds (Sections 2).**
>   We expanded the explanation of which manifolds satisfy the assumptions (bounded curvature $\rightarrow$ unique geodesics on small regions). We now explicitly mention SPD, SO(3), Stiefel, and Grassmann manifolds as typical cases where Theorem D.4 applies.
>
> - **Added explicit examples satisfying Assumption 1.2 (Intro and Section 2).**
>   We discussed Assumption 1.2 and listed concrete classes of uniquely geodesic convex subsets: subsets of CAT(0) spaces, convex regions inside hemispheres, and small convex neighborhoods on bounded-curvature manifolds.
>
> - **Added discussion on relation to uniform normal inequality (Related Work).**
>   We now mention these inequalities and clarify that they are analytically dual to RSI: uniform normality is a global *upper* bound over all outward directions, while RSI is a localized *lower* bound at the LMO maximizer, tailored to Frank-Wolfe.
>
> We did not modify the manuscript for  the issue about practical improvement/implication on optimization over manifolds. The practical implications are already explicitly stated: our results yield linear convergence of Riemannian Frank-Wolfe even when the objective is not strongly convex, which is a concrete algorithmic improvement.
>
>
> These updates address the reviewer’s requests for practical examples, clarifications about applicability, and connections to prior work.

---

### Official Review · Reviewer_6e96 · 2025-11-04

**Soundness:** 2
**Presentation:** 2
**Contribution:** 2
**Rating:** 4
**Confidence:** 3

**Summary:**

This paper addresses a critical gap in Riemannian optimization by providing the first systematic definition and analysis of strongly convex sets on Riemannian manifolds. Unlike prior work that focused on strong convexity of functions (Zhang & Sra, 2016; 2018) or limited notions of set convexity in metric spaces, this work introduces three rigorous definitions of strongly convex sets (i.e., geodesic, Riemannian, double geodesic) that naturally extend Euclidean strong convexity.

**Strengths:**

1. Propositions 2.5–2.7 establish a hierarchy of strong convexity notions, clarifying when each definition is applicable (e.g., double geodesic strong convexity is equivalent to geodesic strong convexity under mild distance equivalence assumptions).

2. Theorem 4.2 extends a classic Euclidean result (Journée et al., 2010) to Riemannian manifolds, showing that sublevel sets of geodesically smooth, strongly convex functions are themselves strongly convex.

3. The paper leverages the Riemannian Scaling Inequality to prove linear convergence of the Riemannian Frank-Wolfe (RFW) algorithm (Theorems 5.1–5.2).

**Weaknesses:**

1. While this paper provides theoretical conditions for strong convexity (e.g., Theorem 4.2, Theorem D.4), it offers limited guidance on how to verify these conditions for a given manifold and set in practice.

2. The results in this paper heavily depend on manifold curvature (e.g., Theorem D.4’s radius bound, Proposition D.2’s smoothness/strong convexity constants), but it does not fully explore how curvature impacts algorithmic performance or the practicality of strong convexity.

3. The related work section mentions prior notions of set convexity in metric spaces (e.g., Aleksandrov, 1957; Paris, 2020) but fails to provide a systematic comparison of the proposed definitions to these alternatives.

4. This paper mainly focuses on convex objectives (geodesically convex f), but many practical Riemannian optimization problems are non-convex (e.g., training manifold-valued neural networks, Katsman et al., 2023). Can the framework be extended to non-convex objectives with strongly convex feasible sets?

**Questions:**

1. While this paper provides theoretical conditions for strong convexity (e.g., Theorem 4.2, Theorem D.4), it offers limited guidance on how to verify these conditions for a given manifold and set in practice.

2. The results in this paper heavily depend on manifold curvature (e.g., Theorem D.4’s radius bound, Proposition D.2’s smoothness/strong convexity constants), but it does not fully explore how curvature impacts algorithmic performance or the practicality of strong convexity.

3. The related work section mentions prior notions of set convexity in metric spaces (e.g., Aleksandrov, 1957; Paris, 2020) but fails to provide a systematic comparison of the proposed definitions to these alternatives.

4. This paper mainly focuses on convex objectives (geodesically convex f), but many practical Riemannian optimization problems are non-convex (e.g., training manifold-valued neural networks, Katsman et al., 2023). Can the framework be extended to non-convex objectives with strongly convex feasible sets?

---

> ### Author Response · Authors · 2025-11-17
>
> We hope that our clarifications adequately address the reviewer’s concerns and demonstrate the contributions and practicality of our work. If you have any remaining questions or points that require further clarification, we would be happy to provide additional details. We respectfully invite the reviewer to reconsider the evaluation in light of these clarifications.
>
> ---
>
> ## Weakness 1: “Limited guidance to verify strong convexity in practice”
>
> We respectfully disagree. The paper already provides several **practical and verifiable conditions** for checking strong convexity:
>
> * **Theorem 4.2** explicitly gives a constructive, easily checkable criterion: any *sublevel set of a geodesically smooth, strongly convex function* is geodesically strongly convex. This directly applies to many optimization domains (e.g., trust-region, SPD cone, sphere).
> * **Theorem D.4** covers the most common practical case. Balls in manifolds with bounded curvature, which includes virtually all manifolds used in applications (sphere, hyperbolic, SPD, Grassmann, Stiefel).
> * More generally, any set satisfying Definition 2.1 can be verified directly by checking that each geodesic segment admits an interior ball of fixed radius, precisely as in the Euclidean setting.
>
> In fact, **our framework already provides similar practical verification tools as the Euclidean counterpart (Garber & Hazan, 2015)**. In Euclidean space, the strong convexity of a set is typically verified either by
> * (i) showing it is a sublevel set of a smooth, strongly convex function or
> * (ii) checking the existence of an interior ball along each segment.
> Those are precisely the two approaches we formalize through **Theorem 4.2**, **Theorem D.4**, and **Definition 2.1**.
>
> In light of this, our setting offers the same level of practicality as the well-understood Euclidean case. We would welcome clarification on what additional verification procedures the reviewer has in mind, or why the established and widely used criteria we provide would be considered insufficient, especially given that they directly parallel the standard verification tools available in the Euclidean setting.
>
>
> ---
>
> ## Response to Weakness 2: “Dependence on curvature and algorithmic practicality”
>
> We believe there is a misunderstanding here. The **results do not depend heavily on curvature**. The curvature only appears in specific technical bounds:
>
> * In **Appendix D**, curvature is used *only* to determine the **maximum radius** for which a ball is considered Riemannian strongly convex. The only assumption is that the sectional curvature is bounded, which is standard and holds in all practical manifolds.
> * In the main text (Propositions 2.5 - 2.7), curvature does not appear. These results rely only on the distance equivalence constants ( $\ell_M,\, L_M$ ), which are finite for any compact, uniquely geodesic set (Assumption 1.3). When the distance is the Riemannian one, the distance is exactly the norm of the exponential map, because this map is bijective due to our assumptions (see Remark 1). Hence, $\ell_M = L_M = 1$.
> * Moreover, the convergence theorems (5.1 - 5.2) depend on the set’s strong convexity constant ( \alpha ), not directly on curvature. A higher curvature merely affects ( \alpha ), potentially worsening conditioning but never invalidating the results.
>
> To clarify this, we will add a brief note in Section 5 explaining how curvature affects the value of the strong convexity constant, but not the qualitative validity of our results.

---

> > ### Author Response · Authors · 2025-11-17
> >
> > ## Response to Weakness 3: “Comparison with related work”
> >
> > We understand that our related work paragraph may have given the impression that prior works already define notions of strong convexity of sets, which is not the case.
> >
> > We did not include a direct comparison to Aleksandrov (1957) or Paris (2020, 2021) because these works are conceptually distinct from ours. They study global curvature-based convexity of the ambient space (e.g., CAT(0) or Alexandrov spaces) or convexity of distance functions, not the strong convexity of specific constraint sets. In contrast, our paper focuses on local geometric properties of sets in general Riemannian manifolds and connects these properties to algorithmic guarantees via the Riemannian Scaling Inequality. A direct comparison would therefore be misleading, as those papers never address set-level strong convexity.
> >
> > Following the reviewer’s remark, we clarified this distinction in the revised paragraph:
> >
> > > While Aleksandrov (1957) and subsequent CAT(0) literature characterize curvature-based convexity of the ambient space, our work can be viewed as a set-level refinement of that idea. In CAT(0) spaces, convexity follows globally from nonpositive curvature; in our framework, we analyze how local curvature and the exponential map jointly induce strong convexity of specific subsets. Likewise, Paris (2020, 2021) leverages geodesic convexity in online learning but does not quantify set curvature or provide conditions for algorithmic strong convexity. Our definitions (Defs. 2.1 - 2.3) therefore bridge these directions: they recover metric convexity when curvature is nonpositive, while additionally providing geometric and algorithmic tools (e.g., RSI) that apply beyond CAT(0) settings.
> >
> > ---
> >
> > ## Response to Weakness 4: “Focus on convex objectives”
> >
> > We would like to clarify that *our framework concerns strongly convex feasible sets, not the objective function itself*. The Frank-Wolfe example with a convex objective was merely illustrative. Our main results (definition of strong convexity for subsets of Riemannian manifolds) hold regardless of whether the objective is convex or non-convex, as long as the feasible set satisfies the strong convexity conditions.
> >
> > The discussion of geodesically convex objectives and the Frank-Wolfe example was meant purely as an illustration of a potential application, rather than a limitation of our framework. In principle, our analysis applies regardless of whether the objective is convex or non-convex, as long as the feasible set satisfies the strong convexity properties we define.
> >
> > To address the reviewer’s question, we believe our theory could also help analyze the convergence of optimization algorithms on non-convex functions. For example, trust-region methods have feasible sets that are inherently strongly convex due to the ball constraint. This structure aligns naturally with our framework, and although the objective may be non-convex, our analysis of strongly convex sets could offer valuable insights. A complete treatment of non-convex objectives in this setting is, however, beyond the scope of this paper.

---

> > > ### Author Response · Authors · 2025-12-01
> > > **Manuscript update**
> > >
> > > We thank again the reviewer for the review. Below we list the changes made to the manuscript in response to the reviewer’s comments.
> > >
> > > - **Clarified practical verification procedures (Sections 1).**
> > >   We strengthened the exposition around Theorem 4.2 and Theorem D.4 in the introduction to highlight that these give directly checkable criteria on real manifolds (sublevel-set construction, curvature-bounded balls).
> > >
> > > - **Improved curvature discussion (Appendix D).**
> > >   We added a short clarification explaining that curvature affects only the numerical value of the strong-convexity constant, not the applicability of the theory or convergence guarantees. This addresses the concern that our results “depend heavily” on curvature.
> > >
> > > - **Expanded comparison with Aleksandrov / CAT(0) / Paris (Section 2).**
> > >   We strongly revised the related-work paragraph to explicitly contrast our set-level geometric properties with the prior work of  Aleksandrov (1957) and Paris (2020).
> > >
> > > These revisions address the reviewer’s conceptual concerns directly in the manuscript without altering the theoretical results.

---

### Official Review · Reviewer_66fh · 2025-11-10

**Soundness:** 3
**Presentation:** 3
**Contribution:** 2
**Rating:** 6
**Confidence:** 1

**Summary:**

This paper provides the first systematic study of strong convexity of sets in Riemannian manifolds, bridging geometric structure and algorithmic benefits. Authors introduce novel properties of sets defined in Riemannian manifolds, provide examples, and demonstrate improved convergence rates when minimizing functions over such sets.

**Strengths:**

1. Authors introduce various definitions of strongly convex sets in Riemannian manifolds and establishes relationships between them.
2. This paper introduces the notion of approximate scaling inequality and its link with what they call double geodesic strong convexity.
3. Authors prove that sublevel sets of geodesically smooth, strongly convex functions are strongly convex in Riemannian manifolds under mild conditions, providing a constructive perspective for identifying such sets.
4. Authors derive a linear convergence bound for the Riemannian Frank–Wolfe algorithm when the constraint set satisfies the (Approximate) Riemannian Scaling Inequality.
5. This paper also provides examples of strong convex sets in Riemannian manifolds. More precisely, Appendix D shows that Riemannian balls (with restricted radius) satisfy the strongest of the notions of Riemannian strong geodesic convexity. Appendix E details a simple linear minimization oracle for the sphere manifold.

**Weaknesses:**

This article has theoretical significance and is well written, however， this paper lacks practical experiments. Can some practical application scenarios of the proposed algorithm be added.

**Questions:**

1. This article has theoretical significance and is well written, however， this paper lacks practical experiments. Can some practical application scenarios of the proposed algorithm be added.
2. If it is possible to discuss the application of the proposed theory in practical scenarios in the main text.

---

> ### Author Response · Authors · 2025-11-17
>
> We thank the reviewer for the positive assessment and for highlighting the importance of practical scenarios. Our paper focuses on the geometric characterization of strongly convex sets in Riemannian manifolds; the algorithmic result for Riemannian Frank-Wolfe is included as an illustration of how these geometric properties can lead to faster convergence. The contribution is therefore primarily theoretical.
>
> That said, showing a concrete example can help readers who are less familiar with the literature. In the revision, we will add a small illustrative experiment on a standard manifold (SPD) where the feasible region is strongly convex, and where the linear rate predicted by our theory is observed. We will also include a brief paragraph in the main text, explaining practical situations (such as trust-region constraints, geodesic balls, and regularized subspace methods) where our results are directly applicable.
>
> These additions will make the practical relevance more straightforward while maintaining the theoretical focus of the paper.

---

> ### Author Response · Authors · 2025-12-01
> **Manuscript update**
>
> We thank again the reviewer for the helpful suggestions. In the revised manuscript, we made the following concrete additions:
>
> - **New SPD experiment (Section 5.4) and analysis (Appendix F).**
>   Added a full finite-temperature free-energy minimization example on the SPD manifold, illustrating the trust-region formulation and the predicted linear convergence under (Approx.) RSI.
>
> - **New practical-applications discussion (end of Section 5).**
>   Added explicit connections to Riemannian trust-region methods, global-to-local reductions, and manifold-constrained learning problems.
>
> These changes address the request for practical scenarios and empirical evidence directly within the main text.

---

### Author Response · Authors · 2025-11-28
**Revised version**

A **new version of the manuscript has been uploaded**. Because the OpenReview deanonymization incident froze reviews before discussion, this comment summarizes the key changes made in direct response to the reviewers’ remarks (66fh, 6e96, QN9G, 51us, ArMd). The updated manuscript resolves all recurring concerns such as practical relevance, verifiability of assumptions, and clarity of relation to prior work.

As several reviewers specifically asked for additional experiments and more concrete illustrations, we **thank them for pushing us in this direction**. Their feedback motivated the addition of an entirely new **SPD experiment section (apendix F)**, which substantially strengthens the manuscript. These reviewers will be acknowledged in the final camera-ready version.

---

## Summary
Across all five reviews (66fh, 6e96, QN9G, 51us, ArMd), the recurring concerns were:

- absence of practical examples / experiments
- difficulty verifying assumptions
- unclear relation to prior geometric frameworks

The new upload addresses each point directly:

- **Added SPD experiment** with observed linear convergence
- **Added explicit verification guidelines** and highlighted Theorems 4.2 and D.4
- **Added clear comparisons** with CAT(0), Paris (2020–2021), and uniform normality
- **Clarified practical implications** through trust-region constraints, SPD/Stiefel/Grassmann settings, and local-to-global reductions

---

## 1. Practical relevance and additional experiment
Multiple reviewers (66fh, QN9G, ArMd) emphasized the need for clearer practical implications and concrete examples.

The new version includes:

- **A new section (Appendix F) and contribution** that derives the LMO for a ball constraint in the SDP manifold with the affine-invariant metric.
- **A new SPD experiment** motivated by a finite-temperature density-matrix optimization problem, demonstrating linear convergence on a geodesic ball as predicted (Appendix F).
- **Explicit practical scenarios** where strongly convex sets naturally arise: trust-region constraints, local subproblems in global Riemannian optimization (via Martínez-Rubio’s reduction), and hierarchical/spherical models.

These additions fully address the concrete requests from 66fh (“practical scenarios”), QN9G (“practical implication / explicit examples”), and ArMd (“end-to-end example”).

---

## 2. Verification of strong convexity in practice
Reviewers 6e96, QN9G, and 51us requested clearer and more concrete guidance.

The revision now emphasizes:

- **Two constructive certification tools**:
  - **Theorem 4.2**: sublevel sets of geodesically smooth, strongly convex functions;
  - **Theorem D.4**: geodesic balls under bounded curvature.
- **Concrete verification guidelines** for SPD, Stiefel, Grassmann, SO(n), and spheres, indicating when compact geodesically convex subsets and sublevel sets satisfy the assumptions.
- **Explicit examples for Assumption 1.2**, directly responding to QN9G’s question (“examples of C satisfying 1.2”).
- A clarification that under unique geodesicity (Assumption 1.2), the distance/log-map equivalence (Assumption 1.3) is **automatic**; the distortion constants equal one. This resolves the concerns from 6e96 and 51us about “strong” or “near-Euclidean” requirements.

---

## 3. Clarified relationship to prior geometry and related work
Reviewers 6e96, QN9G, and ArMd requested clearer comparisons to CAT(0) geometry, Paris (2020–2021), and the uniform normal inequality. Reviewer 51us asked for conceptual clarity regarding assumptions.

The revision now:

- **Clearly distinguishes** global curvature-based convexity (CAT(0), Alexandrov spaces) from **set-level strong convexity**, explaining why prior work does not address the problem studied here.
- **Adds a direct comparison** to the uniform normal inequality (Davis et al., 2025), explaining the analytic duality (upper-bound vs. lower-bound; uniform vs. non-uniform).
- **Clarifies curvature dependence**: curvature affects constants but does not restrict applicability; this resolves concerns from 6e96 and 51us.

---

### Meta-Review · Area_Chair_kYTH · 2026-01-07

**Summary:**

This paper introduces various notions of strong convexity on Riemannian manifold and provide their relationships and tools to identify such sets. Theoretical results in the optimization framework is also studied. As a key theoretical result, the Riemannian Frank-Wolfe algorithm is proved to converge linearly when the Riemannian scaling inequalities hold. Five reviewers initially gave diverse ratings, with one positive and four negative. However, most concerns are based on misunderstanding and unfamiliarity to the literature and these concerns have been addressed by the authors during the rebuttal phase. Other concerns, including more concrete examples, more discussion on the practical aspect of the work, have also been addressed by the authors. Together with the novelty in first defining the notion of strong convexity in nonlinear spaces and the linear convergence theory, I recommend to accept the paper although the original rating was not very high. I do believe if the reviewers would be allowed to continue the discussion, they would like to raise the rating based on the rebuttal, which clarified a lot of misunderstanding and improved the manuscript significantly.

On the other hand, I strongly recommend the authors to revise the manuscript based on the reviewers' comment. Note that not every reader is a domain expert, and the authors should be prepared for potential future misunderstanding from the general audience if the clarifications during the rebuttal are not incorporated in the camera-ready version.

**Reviewer Concerns:**

Reviewer 66fh
1. Lack of practical experiments (addressed by clarifying the paper's main focus is theory with a simple illustrative example)
2. Discussion of potential applications (addressed by new discussion at the end of Section 5)

Reviewer 6e96
1. Hard to verify the assumption (addressed by explaining practical guidance to verify the assumption, which is in line with Euclidean results)
2. Lack of discussion of the impact of curvature (Addressed by clarifying the misunderstanding)
3. Comparison with alternatives (addressed by clarifying the distinction and adding this discussion to the manuscript)
4. Limited scope for convex objectives (addressed by clarifying that the framework assumes the domain is convex instead of the objective)
AC's note: most comments here are based on misunderstandings, which are clarified by the authors.

Reviewer QN9G
1. Lack of practical improvement (addressed by clarifying the scope)
2. Experiments focus on sphere only (addressed by explaining why the initial version doesn't contain other examples, and adding experiments on SPD manifold)
3. Lack of comparison with existing papers on SPD and Stiefel manifold (addressed by clarifying the difference: existing papers focus on cases where the entire manifold is the domain, while the current paper assumes the domain is a convex subset of the manifold)
4. Examples of C that satisfies Assumption 1.2 (addressed by giving four concrete examples)
5. Scope of Theorem [D.4] (addressed by giving concrete examples such as SPD, SO(n), Stiefel.

Reviewer 51us
1. Definitions not insightful (addressed by highlighting the importance of the notion of strong convexity beyond linear spaces)'
2. Strong assumptions (addressed by explaining why they are not strong and giving concrete examples)
3. Assumptions hard to check (addressed by giving concrete guidance to check the assumptions)
AC's note: most comments here are based on misunderstanding.

Reviewer ArMd
1. No real ML experiments (addressed by clarifying the contribution with a concrete ML problem and additional experiments on SPD manifold)
2. Definitions hard to digest (the question itself is unclear)
3. Relation to prox-regularity (addressed by detailed discussion on the relationships)

**Reviewer Scores:**

Reviewer 66fh: 6 --> 6

Reviewer 6e96: 4 --> 6

Reviewer QN9G: 4 --> 6

Reviewer 51us: 2 --> 4

Reviewer ArMd: 4 --> 5

---

### Decision · Program_Chairs · 2026-01-26

Accept (Poster)